# Critical Initialization of Wide and Deep Neural Networks using Partial Jacobians: General Theory and Applications

**Darshil Doshi**[*†]
ddoshi@umd.edu

**Tianyu He**[*†]
tianyuh@umd.edu

**Andrey Gromov** [‡†]
gromovand@meta.com

## Abstract

Deep neural networks are notorious for defying theoretical treatment. However, when the number of parameters in each layer tends to infinity, the network function is a Gaussian process (GP) and quantitatively predictive description is possible. Gaussian approximation allows one to formulate criteria for selecting hyperparameters, such as variances of weights and biases, as well as the learning rate. These criteria rely on the notion of criticality defined for deep neural networks. In this work we describe a new practical way to diagnose criticality. We introduce *partial Jacobians* of a network, defined as derivatives of preactivations in layer $l$ with respect to preactivations in layer $l_0 \leq l$. We derive recurrence relations for the norms of partial Jacobians and utilize these relations to analyze criticality of deep fully connected neural networks with LayerNorm and/or residual connections. We derive and implement a simple and cheap numerical test that allows one to select optimal initialization for a broad class of deep neural networks; containing fully connected, convolutional and normalization layers. Using these tools we show quantitatively that proper stacking of the LayerNorm (applied to preactivations) and residual connections leads to an architecture that is critical for any initialization. Finally, we apply our methods to analyze ResNet and MLP-Mixer architectures; demonstrating the everywhere-critical regime. [4]

## 1 Introduction

When the number of parameters in each layer becomes large, the functional space description of deep neural networks simplifies dramatically. The network function, $f(x)$, in this limit, is a Gaussian process [37, 30] with a kernel – sometimes referred to as neural network Gaussian process (NNGP) kernel [30] – determined by the network architecture and hyperparameters (*e.g* depth, precise choices of layers and the activation functions, as well as the distribution of weights and biases). A similar line of reasoning was earlier developed for recurrent neural networks [36]. Furthermore, for special choices of parameterization and MSE loss function, the training dynamics under gradient descent can be solved exactly in terms of the neural tangent kernel (NTK) [27, 31]. A large body of work was devoted to the calculation of the NNGP kernel and NTK for different architectures, calculation of the finite width corrections to these quantities, and empirical investigation of the training dynamics of wide networks [39, 54, 25, 14, 3, 33, 1, 18, 19, 45, 58, 47, 6, 5, 32, 63, 61, 60, 59, 35, 17, 2, 49, 34].

---

[*]Equal Contributions

[†]Condensed Matter Theory Center & Department of Physics, University of Maryland, College Park, MD 20740

[‡]Meta AI, Menlo Park, CA 94025

[4]Code to reproduce our results is available at https://github.com/ablghtianyi/PartialJac.git

37th Conference on Neural Information Processing Systems (NeurIPS 2023).

One important result that arose from these works is that the network architecture determines the most appropriate initialization of the weights and biases [44, 46, 30]. To state this result, we consider networks with/without LayerNorm [7] and residual connections [21]; the preactivations for which can be defined as follows

$$h_i^{l+1}(x) = \sum_{j=1}^{N_l} w_{ij}^{l+1} \phi(\tilde{h}_j^l(x)) + b_i^{l+1} + \mu h_i^l(x) , \qquad (1)$$

where $\tilde{h}_j^l = \mathrm{LayerNorm}(h_j^l)$ and the parameter $\mu$ controls the strength of residual connections. For the input layer: $h_i^1(x) = \sum_{j=1}^{N_0} w_{ij}^1 x_j + b_i^1$. In the $(l+1)$-th layer, weights $w_{ij}^{l+1} \in \mathbb{R}^{N_{l+1} \times N_l}$ and biases $b_i^{l+1} \in \mathbb{R}^{N_{l+1} \times 1}$ are taken from normal distributions $\mathcal{N}(0, \sigma_w^2/N^l)$ and $\mathcal{N}(0, \sigma_b^2)$, respectively. Hyperparameters $\sigma_w$ and $\sigma_b$ need to be tuned. $\phi(\cdot)$ is the activation function and $x \in \mathbb{R}^{N_0 \times 1}$ is the input. For results discussed in this work, $x$ can be sampled from either a realistic (*i.e.* highly correlated) dataset or a high entropy distribution.

For a network of depth $L$, the network function is given by $f(x) = h^L(x)$. Different network architectures and activation functions, $\phi$, lead to different "optimal" choices of $(\sigma_w, \sigma_b)$. The optimal choice can be understood, using the language of statistical mechanics, as a critical point (or manifold) in the $\sigma_b$–$\sigma_w$ plane. The notion of criticality becomes sharp as the network depth, $L$, becomes large. Criticality ensures that NNGP kernel and the gradients' norm remain $O(L^0)$ as the network gets deeper [45]. Very deep networks will not train unless initialized critically, since the gradients explode or vanish exponentially. Note that high trainability does not imply that the trained model has great performance (test accuracy) after training.

## 1.1 Results

Here we focus on two main results of this work: (i) empirical method to check criticality of a neural network and (ii) an architecture based on layer normalization and residual connections that is critical for *any* initialization. First we introduce the notion of a partial Jacobian.

**Definition 1.1.** Let $h_i^l(x)$ be preactivations of a neural network $f(x)$. The partial Jacobian $J_{ij}^{l_0,l}$ is defined as derivative of preactivations at layer $l$ with respect to preactivations at layer $l_0 \leq l$

$$J_{ij}^{l_0,l}(x) = \frac{\partial h_j^l(x)}{\partial h_i^{l_0}(x)} . \qquad (2)$$

The partial Jacobian is a random matrix with vanishing mean at initialization. Fixing $l_0$ and varying $l$ allows us to study the behavior of Jacobian with depth. On the other hand, varying both $l_0$ and $l$ is necessary to study general networks with non-repeating building blocks. Next, we introduce a deterministic measure of the magnitude of $J_{ij}^{l_0,l}$ — its squared Frobenius norm, averaged over parameter-initializations.

**Definition 1.2.** Let $J_{ij}^{l_0,l}$ be a partial Jacobian of a neural network $f(x)$. Averaged partial Jacobian norm (APJN)[5] is defined as

$$\mathcal{J}^{l_0,l}(x) \equiv \mathbb{E}_\theta \left[ \frac{1}{N_l} \sum_{j=1}^{N_l} \sum_{i=1}^{N_{l_0}} \left( \frac{\partial h_j^l(x)}{\partial h_i^{l_0}(x)} \right)^2 \right] , \qquad (3)$$

where $\mathbb{E}_\theta$ indicates averaging over parameter-initializations.

APJN is a dominant factor in the magnitude of the gradients. In what follows, we show that criticality, studied previously in the literature, occurs when APJN either remains finite, or varies *algebraically* as $l$ becomes large. To prove this we derive the recurrence relation for $\mathcal{J}^{l_0,l}(x)$ in the limit $N_l \to \infty$ and analyze it at large depth. Algebraic behavior of APJN with depth is characterized by an architecture-dependent critical exponent, $\zeta$, so that $\mathcal{J}^{l_0,l}(x) \approx l^{-\zeta}$. Such behavior is familiar from statistical mechanics when a system is tuned to a critical point [10]. Away from criticality, there are two phases:

---

[5]In practice, we compute APJN using a very cheap estimator, viz. Hutchinson's trace estimator[24].

ordered and chaotic. In the ordered phase APJN vanishes exponentially with depth, whereas in the chaotic phase APJN grows exponentially

$$\mathcal{J}^{l_0,l} \approx c_{l_0} e^{\pm \frac{l}{\xi}} \,. \tag{4}$$

Here $\xi$ is the correlation length. It characterizes how fast gradients explode or vanish. (See Section 2.1 for further discussion.)

**Theorem 1.3** (Main result). *Let $f(x)$ be a deep MLP network with Lipschitz continuous activation $\phi(\cdot)$. Assume that LayerNorm is applied to preactivations and there are residual connections with strength $\mu$ acting according to (1). In the sequential limit $(N_1 \to \infty, \dots, N_l \to \infty, \dots, N_{L-1} \to \infty)$, the correlation length with a large $L$ can be written as*

$$\xi = \frac{1}{\left| \log \left[ (1 - \mu^2) \frac{A}{B} + \mu^2 \right] \right|} \,, \tag{5}$$

*where the non-negative constants $A$ and $B$ are given by*

$$A = \sigma_w^2 \mathbb{E}_\theta \left[ \phi'(\tilde{h}_k^{L-2})^2 \right] \,, \qquad\qquad B = \sigma_w^2 \mathbb{E}_\theta \left[ \phi(\tilde{h}_k^{L-2})^2 \right] + \sigma_b^2 \,,$$

*and $\phi'(\cdot)$ is the derivative of $\phi(\cdot)$. We omitted the average over the neuron index $k$ in the infinite width limit, and the result does not depend on $k$.*

*Remark* 1.4. As $\mu$ increases, the dependence on the initialization $(\sigma_b, \sigma_w)$ gets weaker. When $\mu = 1$, the correlation length diverges; and the network is critical for **any** initialization, with $\zeta = O(1)$.

In practice, Theorem 1.3 and Remark 1.4 imply that different choices of initialization bear no effect on trainability of the network provided that LayerNorm and residual connections are arranged as stated.

## 1.2   Related Work

Some of our results were either surmised or obtained in a different form in the literature. We find that LayerNorm ensures that NNGP kernel remains finite at any depth as suggested in the original work of Ba et al. [7]. LayerNorm also alters the criticality of $\mathcal{J}^{l_0,l}(x)$. It was noted in Xu et al. [57] that LayerNorm (applied to preactivations) regularizes the backward pass. We formalize this observation by showing that LayerNorm (applied to preactivations) dramatically enhances correlation length (which is not the case for LayerNorm applied to activations). This can be seen from Theorem 1.3, setting $\mu = 0$. When residual connections of strength 1 are combined with erf (or any other erf-like activation function, e.g. tanh), the neural network enters a *subcritical* phase with enhanced correlation length (see 4.3). A version of this result was discussed in Yang and Schoenholz [62]. When residual connections are introduced on top of LayerNorm, the correlation length $\xi$ is further increased. If residual connections have strength $\mu = 1$ the network enters a critical phase for *any* initialization. Importance of correct ordering of LayerNorm, residual connections and attention layers was discussed in Xiong et al. [56]. Benefit of combining BatchNorm and residual connections was mentioned in [12]. Several architectures with the same order of GroupNorm and residual connections were investigated in Yu et al. [64]. Existing initialization schemes such as He initialization[6] [20], Fix-up [65] and ReZero [8] are special cases of our method. They conform to the notion of criticality using APJN, as defined in our work.

The partial Jacobian has been used to study generalization bounds in Arora et al. [4]. The Jacobian norm (*i.e.* $||J_{ij}^{0,l}||^2$) of trained feed-forward neural networks was studied in Novak et al. [38], where it was correlated with generalization. Partial Jacobians with $l_0 = l - 1$ were studied in the context of RNNs [11, 9], referred to as *state-to-state* Jacobians. Furthermore, the relation between Partial Jacobians and Lyapunov exponents, as well as their impact on trainability, have been explored for RNNs [16, 15].

As the aspect ratio $(L/N)$ of the network approaches 1, the finite width corrections to the Jacobian become more prominent. On the other hand, even with a small aspect ratio, the effect of the spectral density of Jacobian becomes important as the depth $L$ becomes *very* large. Pennington et al. [43] studied the spectrum of the input-output Jacobian for MLPs. Xiao et al. [54] extended the analysis to CNNs, showing that *very* deep vanilla CNNs can be trained by achieving "dynamical isometry".

---

[6]He initialization is critical only for ReLU networks.

## 2 Recurrence Relations

Here we derive the infinite width recurrence relations for the APJN and the NNGP kernel. We use Lemma 2.2 to derive NNGP kernel recurrence relation, and leverage that to get the recurrence relation for APJN. Fixed point analyses of these relations help us define critical line and point. We start with the vanilla MLP with no LayerNorm and $\mu = 0$. Results with LayerNorm and/or residual connections, as well as modern architectures are presented in the following sections. (We refer the reader to Appendices D to F for proofs and detailed application of this recipe. Appendices I to K contain the calculations and results for various activation functions.)

**Definition 2.1.** We define averaged covariance of preactivations as follows

$$\mathcal{K}^l(x, x') = \mathbb{E}_\theta \left[ \frac{1}{N_l} \sum_{i=1}^{N_l} h_i^l(x) h_i^l(x') \right] . \tag{6}$$

**Lemma 2.2.** *When $N_l \to \infty$ for $l = 1, \ldots, L-1$ sequentially, the expectation value over parameter initializations for a general function of preactivations: $\mathcal{O}(h^l(x))$, can be expressed as the averaging over the Gaussian process $h^l(x)$ with covariance $\mathcal{K}^l(x, x')$.*

$$\mathbb{E}_\theta \left[ \mathcal{O}(h_i^l(x)) \right] = \frac{1}{\sqrt{2\pi \mathcal{K}^l(x, x)}} \int dh^l \mathcal{O}(h_i^l(x)) e^{-\frac{(h_i^l(x))^2}{2\mathcal{K}^l(x,x)}} . \tag{7}$$

This result has been established in Lee et al. [30]. Note that the density in (7) only depends on the diagonal part of the covariance matrix, $\mathcal{K}^l(x, x)$. We will refer to $\mathcal{K}^l(x, x)$ as NNGP kernel.

*Remark* 2.3. In the sequential infinite width limit the means appearing in (9)-(11) are self-averaging and, therefore, deterministic. They converge in distribution to their averages over parameterizations.

$$\frac{1}{N_l} \sum_{i=1}^{N_l} \phi(h_i^l)^2 \xrightarrow{N_{l'\leq l-1}\to\infty} \mathbb{E}_\theta \left[ \frac{1}{N_l} \sum_{i=1}^{N_l} \phi(h_i^l)^2 \right] . \tag{8}$$

When performing analytic calculations we use the infinite width convention; whereas in our finite-width experiments we explicitly average over initializations of $\theta^l$.

**Theorem 2.4.** *With 2.2, in the infinite width limit, the NNGP kernel $\mathcal{K}^{l+1}(x, x)$ is deterministic, and can be determined recursively via*

$$\mathcal{K}^{l+1}(x, x) = \sigma_w^2 \mathbb{E}_\theta \left[ \frac{1}{N_l} \sum_{i=1}^{N_l} \phi(h_i^l(x))^2 \right] + \sigma_b^2 . \tag{9}$$

**Theorem 2.5.** *Let $f(x)$ be an MLP network with a Lipschitz continuous activation function $\phi(x)$. In the infinite width limit, APJN $\mathcal{J}^{l_0,l+1}(x)$ is deterministic and satisfies a recurrence relation*

$$\mathcal{J}^{l_0,l+1}(x) = \chi_{\mathcal{J}}^l \mathcal{J}^{l_0,l}(x) , \tag{10}$$

*where the factor $\chi_{\mathcal{J}}^l$ is given by*

$$\chi_{\mathcal{J}}^l = \sigma_w^2 \mathbb{E}_\theta \left[ \frac{\sigma_w^2}{N_l} \sum_{i=1}^{N_l} \left( \phi'(h_i^l(x)) \right)^2 \right] . \tag{11}$$

2.4 is due to Neal [37] and Lee et al. [30]. 2.5 is new and is valid only in the limit of infinite width. The proof is in D. We will drop the explicit dependence on $x$ to improve readability.

The expectation values that appear in (9)-(11) are evaluated using (7). When the integrals can be taken analytically, they lead to explicit equations for the critical lines and/or the critical points. Details of these calculations as well as the derivation of (9)-(11) can be found in the D. A subtlety emerges in (10) when $l_0 = 0$, where a correction of the order $O(N_0^{-1})$ arises for non-scale invariant activation functions. This subtlety is discussed in the D.

When the depth of the network becomes large, the $l$-dependence of the expectation values that appear in (6), (11) saturate to a (possibly infinite) constant value; which means that $\mathcal{K}^l$, $\mathcal{J}^{l_0,l}$ and $\chi_{\mathcal{J}}^l$ have

reached a fixed point. We denote the corresponding quantities as $\mathcal{K}^\star, \mathcal{J}^{l_0,\star}, \chi_{\mathcal{J}}^\star$. The existence of a fixed point is not obvious and should be checked on a case by case basis. Fixed point analysis for $\mathcal{K}^l$ was done in Poole et al. [44] for bounded activation functions and in Roberts et al. [45] for the general case. The stability is formulated in terms of

$$\chi_{\mathcal{K}}^\star = \left.\frac{\partial \mathcal{K}^{l+1}}{\partial \mathcal{K}^l}\right|_{\mathcal{K}^l = \mathcal{K}^\star}. \tag{12}$$

The norm of preactivations remains finite (or behaves algebraically) when $\chi_{\mathcal{K}}^\star = 1$.

Eq. (10) nicely expresses $\mathcal{J}^{l_0,l+1}$ as a linear function of $\mathcal{J}^{l_0,l}$. The behaviour of $\mathcal{J}^{l_0,l+1}$ at large $l$ is determined by $\chi_{\mathcal{J}}^l$. When $\chi_{\mathcal{J}}^l > 1$ partial Jacobians diverge exponentially, while for $\chi_{\mathcal{J}}^l < 1$ partial Jacobians vanish exponentially. Neural networks are trainable only up to a certain depth when initialized $O(1)$ away from criticality, which is determined by the equation

$$\chi_{\mathcal{J}}^\star = 1. \tag{13}$$

Eq. (13) is an implicit equation on $\sigma_b, \sigma_w$ and generally outputs a critical line in $\sigma_b$–$\sigma_w$ plane. The parameter $\chi_{\mathcal{J}}^\star$ has to be calculated on a case-by-case basis using either (11) or the method presented in the next section. Everywhere on the critical line, $\mathcal{J}^{l_0,l}$ saturates to a constant or behaves algebraically.

When the condition $\chi_{\mathcal{K}}^\star = 1$ is added, we are left with a critical point[7]. This analysis of criticality at infinite width agrees with Roberts et al. [45], where $\chi_\perp$ is to be identified with $\chi_{\mathcal{J}}^\star$; and Schoenholz et al. [46], Martens et al. [34], where their analysis based on the equivalent $\chi_1$ or $C'(1)$ only works for bounded activation functions. In particular, condition (13) together with $\chi_{\mathcal{K}}^\star = 1$ ensures that NTK is $O(1)$ at initialization.

## 2.1 Empirical Diagnostic of Criticality

APJN $\mathcal{J}^{l_0,l}$ provides a clear practical way to diagnose whether the network is critical or not. Proper choice of $l_0$ and $l$ allows us to minimize the non-universal effects and cleanly extract $\chi_{\mathcal{J}}^\star$.

Recurrence relation (10), supplemented with the initial condition $\mathcal{J}^{l_0,l_0+1} = \chi_{\mathcal{J}}^{l_0}$, can be formally solved as

$$\mathcal{J}^{l_0,l} = \prod_{\ell=l_0}^{l-1} \chi_{\mathcal{J}}^\ell. \tag{14}$$

We would like to obtain an estimate of $\chi_{\mathcal{J}}^\star$ as accurately as possible. To that end, imagine that for some $l' > l_0$ the fixed point has been essentially reached and $\chi_{\mathcal{J}}^{l'} \approx \chi_{\mathcal{J}}^\star$. Then the APJN

$$\mathcal{J}^{l_0,l} = (\chi_{\mathcal{J}}^\star)^{l-l'-1} \cdot \prod_{\ell=l_0}^{l'} \chi_{\mathcal{J}}^\ell \tag{15}$$

depends on the details of how the critical point is approached; which are encoded in the last factor.

**Proposition 2.6.** *If the network is homogeneous,* i.e., *consists of (possibly complex) blocks of layers, periodically repeated $L$ times; then the penultimate APJN provides an accurate estimate of $\chi_{\mathcal{J}}^\star$:*

$$\left.\mathcal{J}^{L-2,L-1}\right|_{L\to\infty} = \chi_{\mathcal{J}}^\star. \tag{16}$$

*This is a direct consequence of combining (9) and (11) as $L$ goes to infinity. See 1 for numerical justification.*

2.6 is the central result of this section and will be heavily used in the remainder of this work.

Note that for deep networks, away from criticality, APJN takes the form

$$\mathcal{J}^{l_0,l} \approx c_{l_0} e^{\pm \frac{l}{\xi}}, \qquad \xi = |\log \chi_{\mathcal{J}}^\star|^{-1}, \tag{17}$$

where $c_{l_0}$ is a non-universal constant that depends on $l_0$. If the sign in (17) is positive ($\chi_{\mathcal{J}}^\star > 1$) the network is in the *chaotic phase*, while when the sign is negative ($\chi_{\mathcal{J}}^\star < 1$) the network is in the

---

[7]Scale-invariant activation functions are more forgiving: away from the critical point, $\mathcal{K}^l$ scales algebraically with $l$.

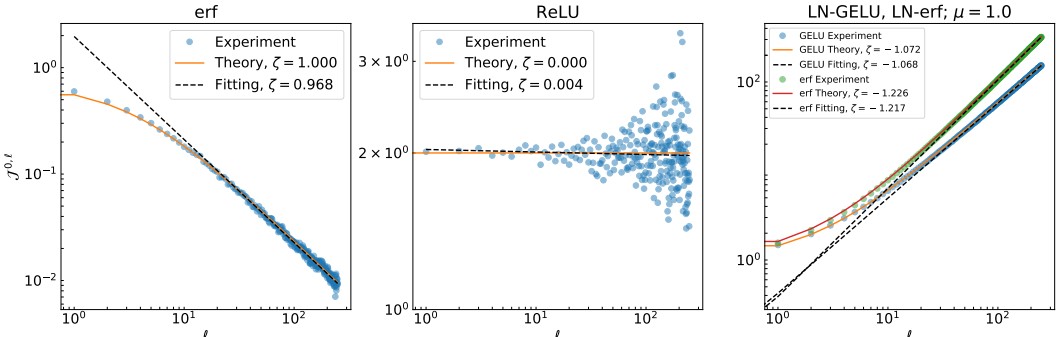

Figure 1: log–log plots of partial Jacobian $\mathcal{J}^{0,l}$ vs. $l$. From left to right: (1)erf, (2)ReLU, (3)erf and GELU with LayerNorm applied to preactivations and residual connections of strength 1. The fluctuations get larger towards the output because the aspect ratio (*i.e.* $l/N_l$) approaches $1/4$.

*ordered phase.* $\xi$ has the meaning of correlation length: on the depth scale of approximately $k\xi$ the gradients remain appreciable, and hence the network with the depth of $\approx k\xi$ will train.

We used (16) to map out the $\sigma_b$–$\sigma_w$ phase diagrams of various MLP architectures. The location of the critical line agrees remarkably well with our infinite width calculations. Results are presented in Fig. 2.

At criticality, $\chi^\star_{\mathcal{J}} = 1$ and the correlation length diverges; indicating that gradients can propagate arbitrarily far. A more careful analysis of non-linear corrections shows that APJN can exhibit algebraic behavior with depth and can still vanish in the infinite depth limit, but much slower than the ordered phase.

## 2.2 Scaling at a Critical Point

At criticality $\chi^l_{\mathcal{J}}$ saturates to a fixed value $\chi^\star_{\mathcal{J}} = 1$. If we are interested in $\mathcal{J}^{l_0,l}$ with $l - l_0 = O(L)$ then it is essential to know how exactly $\chi^l_{\mathcal{J}}$ approaches 1.

**Theorem 2.7.** *Assume that deep neural network $f(x)$ is initialized critically. Then $l \to \infty$ asymptotics of APJN is given by*

$$\mathcal{J}^{l_0,l}(x) = O(l^{-\zeta}),\tag{18}$$

where $\zeta$ is the critical exponent Roberts et al. [45], see Appendix G for further details.

Critical exponents can be determined analytically in the limit of infinite width. Note that $\chi^l_{\mathcal{J}}$, given by (11), depends on $\mathcal{K}^l$ by virtue of (7). Consequently, Eqs. (9)-(11) are coupled through non-linear (in $\mathcal{K}^l$ and $\mathcal{J}^{l_0,l}$) terms. These non-linear corrections are absent for any scale-invariant activation function, but appear for other activation functions.

We checked the scaling empirically by plotting $\mathcal{J}^{0,l}$ vs. $l$ in a log–log plot and fitting the slope. These results are presented in Fig.1, and are in excellent agreement with infinite width calculation.

## 3 Layer Normalization

The fact that critical initialization is concentrated on a single point $(\sigma^\star_w, \sigma^\star_b)$ may appear unsettling because great care must be taken to initialize the network critically. The situation can be substantially improved by utilizing the normalization techniques known as LayerNorm [7] and GroupNorm [52]. Our results apply to GroupNorm verbatim in the case when the number of groups is much smaller than the width. LayerNorm can act either on preactivations or on activations (discussed in the Appendix D). Depending on this choice, criticality will occur on different critical *lines* in $\sigma_b$–$\sigma_w$ plane. When LayerNorm is applied to *preactivations* the correlation length is enhanced, allowing for training much deeper networks even far away from criticality.

The LayerNorm applied to preactivations takes the following form

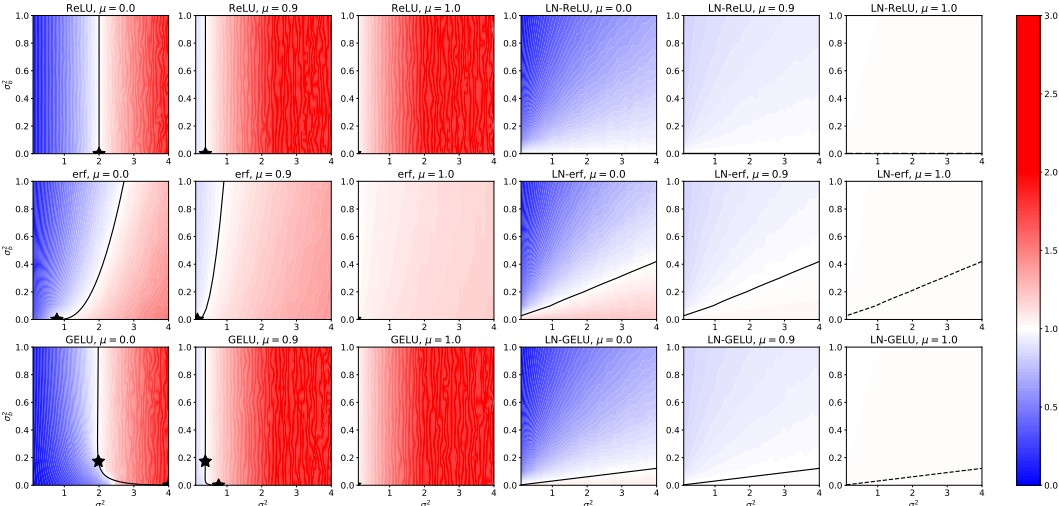

Figure 2: $\chi^\star_\mathcal{J}$ empirical phase diagrams for an MLP with $L = 50, N = 500$. The solid lines indicate the critical lines obtained through infinite width limit calculations, and the stars indicate the critical points. The dotted lines in the rightmost column correspond to the critical lines for $\mu < 1$ case. For networks with LayerNorm and $\mu = 1$, $\chi^\star_\mathcal{J} = 1$ holds on the entire $\sigma_b$–$\sigma_w$ plane. We also note that for erf activation, the case $\mu = 1$ *without* LayerNorm is subcritical and has a large correlation length.

**Definition 3.1** (Normalized preactivations)**.**

$$\tilde{h}^l_i = \frac{h^l_i - \mathbb{E}[h^l]}{\sqrt{\mathbb{E}[(h^l)^2] - \mathbb{E}[h^l]^2}} \xrightarrow{N_l \to \infty} \frac{1}{\sqrt{\mathcal{K}^l}} h^l_i \,, \tag{19}$$

where we have introduced $\mathbb{E}[h^l] = (\sum_{i=1}^{N_l} h^l_i)/N_l$. In the limit of infinite width $\mathbb{E}[h^l] = 0$ and $\mathbb{E}[(h^l)^2] = \mathcal{K}^l$, defined according to (6).

Normalized preactivations, $\tilde{h}^l_i$, are distributed according to $\mathcal{N}(0, 1)$ for all $l, \sigma_w, \sigma_b$. The norms are, therefore, *always* finite and the condition $\chi^\star_\mathcal{K} = 1$ is trivially satisfied. This results in a critical line rather than a critical point.

The recurrence relations (9)-(11) for the NNGP and partial Jacobians are only slightly modified

$$\mathcal{K}^{l+1} = \sigma_w^2 \mathbb{E}_\theta \left[ \frac{1}{N_l} \sum_{i=1}^{N_l} \phi(\tilde{h}^l_i)^2 \right] + \sigma_b^2 \qquad \chi^l_\mathcal{J} = \frac{\sigma_w^2}{\mathcal{K}^l} \mathbb{E}_\theta \left[ \frac{1}{N_l} \sum_{i=1}^{N_l} \phi'(\tilde{h}^l_i)^2 \right] \,. \tag{20}$$

Assuming that the value of $\chi^l_\mathcal{J}$ at the fixed point is $\chi^\star_\mathcal{J}$, the network is critical when (13) holds.

$\chi^l_\mathcal{J}$ (20) is depth independent and changes slowly with $\sigma_w$ and $\sigma_b$. Thus, $\chi^\star_\mathcal{J}$ remains close to 1 for a wider range of hyperparameters. Consequently, the correlation length is *large* even away from criticality, leading to a much higher trainability of deep networks.

## 4 Residual (Skip) Connections

Adding residual connections between the network layers is a widely used technique to facilitate the training of deep networks. Originally introduced [21] in the context of convolutional neural networks [29] (CNNs) for image recognition, residual connections have since been used in a variety of networks architectures and tasks [50, 13].

Consider (1) with non-zero $\mu$ and without LayerNorm layers. Then the recurrence relations (9)-(11) for the NNGP kernel and $\chi^l_\mathcal{J}$ are modified as follows

$$\mathcal{K}^{l+1} = \sigma_w^2 \mathbb{E}_\theta \left[ \frac{1}{N_l} \sum_{j=1}^{N_l} \phi(h^l_j)^2 \right] + \sigma_b^2 + \mu^2 \mathcal{K}^l, \quad \chi^l_\mathcal{J} = \sigma_w^2 \mathbb{E}_\theta \left[ \frac{1}{N_l} \sum_{k=1}^{N_l} \phi'(h^l_k)^2 \right] + \mu^2 \,. \tag{21}$$

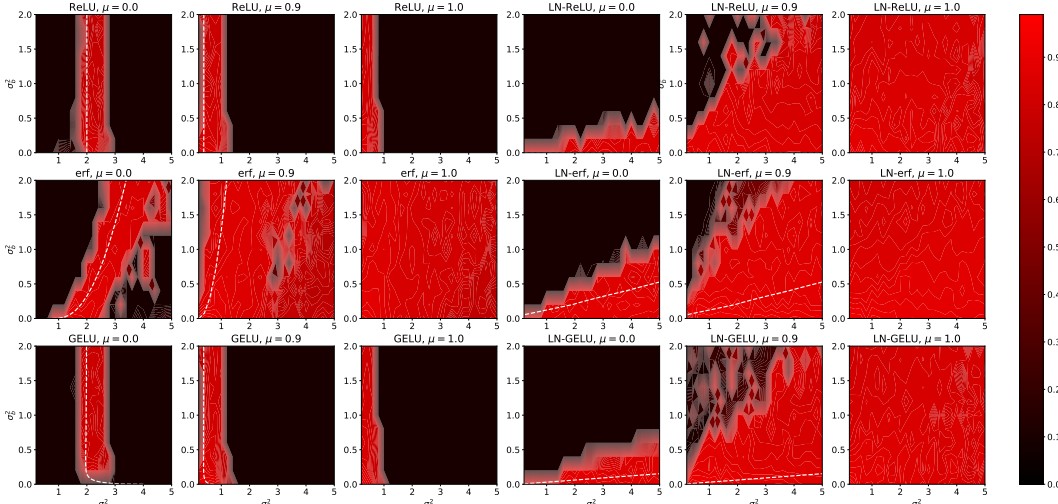

Figure 3: Trainability (Training Accuracy) of deep MLP ($N_l = 500$, $L = 50$) on FashonMNIST. The combination of LayerNorm and $\mu = 1$ makes the network everywhere-trainable. The subcritical case of erf activation *without* LayerNorm, and $\mu = 1$ also has enhanced trainability. The dashed white lines denote the (analytical) critical lines.

*Remark* 4.1. When $\mu < 1$, the fixed point value of NNGP kernel is scaled by $(1 - \mu^2)^{-1}$. For $\mu = 1$, the critical point is formally at $(0, 0)$.

*Remark* 4.2. For $\mu = 1$, (21) implies that $\chi^l_{\mathcal{J}} \geq 1$, where the equality holds on the $\sigma_w = 0$ axis. Consequently, APJN exponentially diverges as a function of depth $l$ for all $\sigma_w > 0$. In this case, $\sigma_w$ needs to be taken sufficiently close to 0 to ensure trainability at large depths.

When $\mu < 1$, residual connections amplify the chaotic phase and decrease the correlation length away from criticality for unbounded activation functions.

Solving the recurrence relations (21) for $\mathrm{erf}$ activation, we find an effect observed in Yang and Schoenholz [62] for $\tanh$ activation. They noted that $\tanh$-like MLP networks with skip connections "hover over the edge of chaos". We quantify their observation as follows.

**Theorem 4.3.** *Let $f(x)$ be a deep MLP network with erf activation function and residual connections of strength $\mu = 1$. Then in the limit $N_l \to \infty$*

- *The NNGP kernel $K^l$ linearly diverges with depth $l$.*

- *$\chi^l_{\mathcal{J}}$ approaches 1 from above (Fig. 2) : $\chi^l_{\mathcal{J}} \approx 1 + \tilde{c}/\sqrt{l}$, where $\tilde{c} = 2\sigma_w^2/(\pi\sqrt{\sigma_w^2 + \sigma_b^2})$ is a non-universal constant. Consequently, APJN diverges as a stretched exponential : $\mathcal{J}^{l_0,l} = O(e^{\sqrt{\frac{l}{\lambda}}})$, where $\lambda = 1/(4\tilde{c}^2)$ is the new length scale.*

We will refer to this case as *subcritical*. Although $\chi^\star_{\mathcal{J}}$ reaches 1, the APJN still diverges with depth faster than any power law. The growth is controlled by the new scale $\lambda$. To control the gradient we would like to make $\lambda$ large, which can be accomplished by decreasing $\sigma_w$. In this case, the trainability is enhanced (see Fig. 3). Similar results hold for $\tanh$ activation function [62], however in that case there is no explicit expression for $\tilde{c}$.

## 5 Residual Connections + LayerNorm

In practice, it is common to use a combination of residual connections and LayerNorm. Using (1), the recurrence relations (9)-(11) for NNGP and partial Jacobians are modified as follows

$$\mathcal{K}^{l+1} = \sigma_w^2 \mathbb{E}_\theta \left[ \frac{1}{N_l} \sum_{j=1}^{N_l} \phi(\tilde{h}_j^l)^2 \right] + \sigma_b^2 + \mu^2 \mathcal{K}^l, \quad \chi^l_{\mathcal{J}} = \frac{\sigma_w^2}{\mathcal{K}^l} \mathbb{E}_\theta \left[ \frac{1}{N_l} \sum_{k=1}^{N_l} \phi'(\tilde{h}_k^l)^2 \right] + \mu^2. \quad (22)$$

*Remark* 5.1. For $\mu < 1$, (22) implies that the fixed point value of NNGP kernel is scaled by $1 - \mu^2$. Moreover, residual connections do not shift the phase boundary. The interference between residual connections and LayerNorm brings $\chi^l_{\mathcal{J}}$ closer to 1 on the entire $\sigma_b$–$\sigma_w$ plane (as can be seen from Fig. 2). Therefore the correlation length $\xi$ is improved in both the phases, allowing for training of deeper networks. At criticality, Jacobians linearly diverge with depth.

As was mentioned before, the combination of LayerNorm and residual connections dramatically enhances correlation length, leading to a more stable architecture. This observation is formalized by 1.3. The proof leverages the solutions of (22) close to the fixed point, and is fleshed out in Appendix F.

*Remark* 5.2. When $\mu = 1$, the correlation length diverges for *any* initialization.

5.2 provides an alternative perspective on architecture design. On the one hand, one can use (16) to initialize a given network at criticality. Alternatively, one can use a combination of residual connections and LayerNorm to ensure that the network will train well, irrespective of the initialization.

*Remark* 5.3. When $\mu = 1$, the condition $\chi^\star_{\mathcal{J}} = 1$ holds on the entire $\sigma_b - \sigma_w$ and for any activation function $\phi$ (see Fig. 2). NNGP kernel diverges linearly, while APJN diverges algebraically with the critical exponent of $\zeta = O(1)$. The exact value of the critical exponent depends on the activation function and the ratio $\sigma_b/\sigma_w$. The trainability is dramatically enhanced, as shown in Fig. 3.

*Remark* 5.4. Networks with BatchNorm [26], used in conjunction with residual connection of strength $\mu = 1$, also enjoy this *everywhere criticality* and enhanced trainability [63, 23] (See Appendix B).

## 6   Modern Architectures

**ResNet110**   ResNet is one of the most widely used architectures in computer vision[21]. Figure 4 shows the results for ResNetV2 [22]; with BatchNorm replaced with LayerNorm. (See Appendix B for BatchNorm results and discussions.)

At $\mu = 1$, $\sigma_w^2 - \sigma_b^2$ phase diagram shows everywhere-criticality, as expected from our theory. For $\sigma_b^2 = 0$, $\sigma_w^2 - \mu$ phase diagram gets closer to criticality as we increase $\mu$, which results in better training at higher $\mu$.

Additionally, networks with larger $\mu$ enjoy better trainability due to the suppression of finite width corrections to our theory (decreased effective $L/N$).

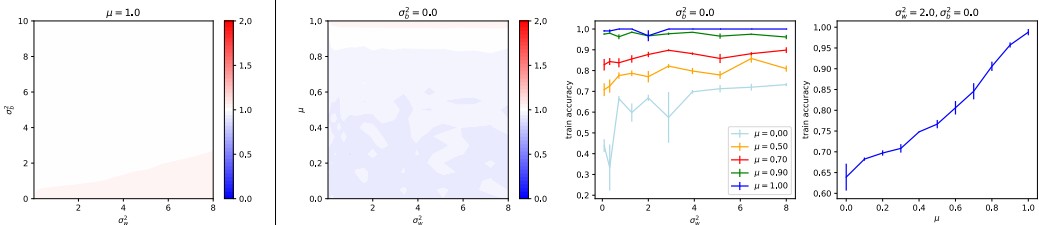

Figure 4: **ResNet110(LayerNorm)**: Left to right: (1) $\chi^\star_{\mathcal{J}}$ phase diagram w.r.t. $(\sigma_w^2, \sigma_b^2)$ at $\mu = 1$. The network is everywhere-critical in this case. (2) $\chi^\star_{\mathcal{J}}$ phase diagram w.r.t. $(\sigma_w^2, \mu)$. (3) Training accuracy w.r.t. $\sigma_w^2$, for different values of $\mu$. Trainability improves with with higher $\mu$, as the network gets closer to everywhere-critical phase. For $\mu = 0$ and small $\sigma_w^2$ the network is in the ordered phase, and the output of every block is almost zero – this explains the poor trainability. (4) Training accuracy w.r.t. $\mu$ at $(\sigma_w^2, \sigma_b^2) = (2, 0)$. We see a monotonic improvement in trainability.

**MLP-Mixer**   MLP-Mixer architecture is a recent example of MLP approach to computer vision [48]. It can be analyzed using the tools presented above. Fig. 5 shows the phase and training diagrams. Further details can be found in the SM. Note that the original architecture uses $\mu = 1.0$.

Test accuracies for figures 4, 5 follow similar trends as train accuracies. (See Appendices B, H).

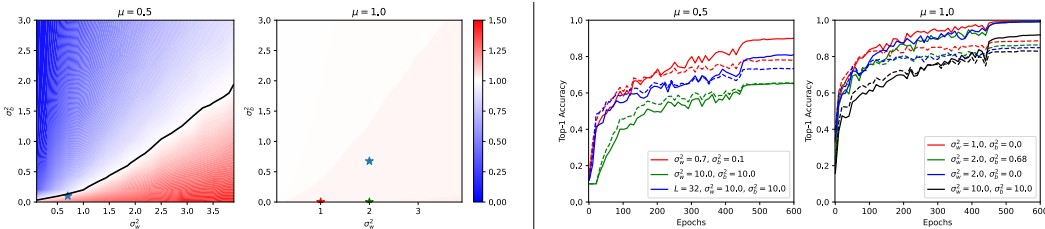

Figure 5: **MLP-Mixer**. Left to right: (1)(2) $\mu = 0.5, 1.0$ phase diagrams ($\chi^\star_{\mathcal{J}}$). The network becomes everywhere-critical when $\mu = 1$. The solid black line indicates the empirical phase boundary. Stars denote points we selected to train on CIFAR-10. (3)(4) $\mu = 0.5, 1.0$ MLP-Mixer training curves. All, but one, networks are $L = 100$ blocks deep. We see that as we increase $\mu$ from $0.5$ to $1$, the trainability of all networks increases, and they are less sensitive to initialization.

# 7    Conclusions

We have introduced partial Jacobians and their averaged norms as tools to analyze the propagation of gradients through deep neural networks at initialization. Using APJN evaluated close to the output, $\mathcal{J}^{L-2,L-1} \approx \chi^\star_{\mathcal{J}}$, we have introduced a very cheap and simple empirical test for criticality. We have also shown that criticality formulated in terms of partial Jacobians is equivalent to criticality studied previously in literature [44, 45, 34]. Additionally, APJN can be utilized to quantify criticality in inhomogeneous (*i.e.* no periodic stacking of blocks) networks [23].

We have investigated homogeneous architectures that include fully-connected layers, normalization layers and residual connections. In the limit of infinite width, we showed that (i) in the presence of LayerNorm, the critical point generally becomes a critical line, making the initialization problem much easier, (ii) LayerNorm applied to preactivations enhances correlation length leading to improved trainability, (iii) combination of $\mu = 1$ residual connections and erf activation function enhances correlation length driving the network to a *subcritical* phase with APJN growing according to a stretched exponential law, (iv) combination of residual connections and LayerNorm drastically increases correlation length leading to improved trainability, (v) when $\mu = 1$ and LayerNorm is applied to preactivations *the network is critical on the entire $\sigma_b$–$\sigma_w$ plane.*

We have considered examples of modern architectures: ResNet and MLP-Mixer. We showed that at $\mu = 1$, they are critical everywhere, due to the interaction between LayerNorm and residual connections. We have studied ResNet at different $\mu$'s, showing that higher $\mu$ enjoys better trainability, due to improved initialization and suppressed effective $L/N$ corrections.

We have empirically demonstrated that deep (100 blocks) MLP-Mixer with $\mu = 1$ trains well for various initializations. In comparison, for $\mu = 0.5$, it only trains well close to the critical line.

Our work shows that an architecture can be designed to have a large correlation length leading to a guaranteed trainability with SGD for any initialization scheme.

# 8    Limitations and Future Works

While our method is empirically applicable to Transformer architectures (see Appendix C for phase diagrams), there remains a notable absence of robust theoretical foundations for initializing Transformers. We aim to extend our theory to the attention mechanism and determine the optimal way to initialize Transformer architectures.

Another challenge arises from network topology. For graph neural networks, the generalization of our methods remains ambiguous. We aspire to elucidate this in future research.

Lastly, we want to scale our experiments to larger image datasets as well as language tasks, with the hope that our methods could help people train large models more efficiently.

## Acknowledgments and Disclosure of Funding

We thank T. Can, G. Gur-Ari, B. Hanin, D. Roberts and S. Yaida for comments on the manuscript. A.G.'s work at the University of Maryland was supported in part by NSF CAREER Award DMR-2045181, Sloan Foundation and the Laboratory for Physical Sciences through the Condensed Matter Theory Center.

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

# A  Experimental Details

We implemented our methods using PyTorch [42] hooks and an efficient Jacobian approximate algorithm [24].

Figure 1: We generated MNIST-like inputs, where all elements are sampled from the Gaussian distribution $\mathcal{N}(0, 1)$. $\mathcal{J}^{0,l}$ data was averaged over 100 different parameter-initializations. Networks were initialized width $N_l = 1000$. For erf plot we initialized at critical point $(\sigma_w, \sigma_b) = (\sqrt{\frac{\pi}{4}}, 0)$, used depth $L = 250$ and the fitting was done with data points collected at depth $l > 100$; for ReLU plot we initialized at critical point $(\sigma_w, \sigma_b) = (\sqrt{2}, 0)$, used depth $L = 100$ and the fitting was done with all data points; for the $\mu = 1$ Pre-LN plot, we initialized both networks at $(\sigma_w, \sigma_b) = (\sqrt{2}, 0)$, used depth $L = 250$ and the fitting was done with $l > 100$ data points.

Figure 2: All the phase diagrams were plotted using $\chi_{\mathcal{J}}^{L-1}$ generated from networks with $L = 50$ and $N_l = 500$. We used hooks to obtain the gradients that go into calculating $\chi_{\mathcal{J}}^{L-1}$. $\chi_{\mathcal{J}}^{L-1}$ data was averaged over 100 different parameter-initializations. Inputs were generated from a normal Gaussian distribution and have dimension $28 \times 28$. Generating the data for the figure took approximately 2 days on Google Colab Pro (single Tesla P100 GPU).

Figure 3: In all cases, networks are trained for 10 epochs using stochastic gradient descent with CrossEntropy loss. We used the Fashion MNIST dataset [53]. All networks had depth $L = 50$ and width $N_l = 500$. The learning rates were logarithmically sampled within $(10^{-5}, 1)$. Generating the data for the figure took approximately 12 days on Google Colab Pro (single Tesla P100 GPU).

Figure 4: (1)We made the $\sigma_w^2 - \sigma_b^2$ phase diagram for ResNet110(LayerNorm) by averaging over 100 different parameter-initializations. The $\sigma_w^2 - \mu$ phase diagram was made by averaging over 200 parameters initialization. (3)(4) We used SGD with momentum$= 0.9$ and batch size 128. For selecting the learning rate we ran a grid-search over $0.001, 0.005, 0.01, 0.02, 0.5$ for 10 epochs; with weight decay $\lambda = 10^{-4}$. All models were trained for 50 epochs and averaged over 3 random seeds. It takes 6 GPU days in total on a single NVIDIA RTX 3090 GPU.

Figure 5: (1)(2)We made the phase diagram for MLP-Mixer with 30 blocks and averaged over 100 different parameter-initializations. (3)(4)We used network with $L = 100$, patch size $4 \times 4$, hidden size $C = 128$, two MLP dimensions $N_{tm} = N_{cm} = 256$. The $L = 32$ point has doubled widths. All networks have 10 million parameters. Notice that for all Mixer Layers we used NTK initialization. We trained all cases on CIFAR-10 dataset using vanilla SGD paired with CSE. Batch size bs $= 256$, weight decay $\lambda = 10^{-4}$ was selected from $\{10^{-5}, 10^{-4}\}$, mixup rate $\alpha = 0.8$ was selected from $\{0.4, 0.8\}$. We also used RandAgument and horizontal flip with default settings in PyTorch. For all cases we searched learning rates within $\{0.005, 0.01, 0.05, 0.1, 0.2, 0.5\}$. We also tried a linear warm-up schedule for first 3000 iterations, but we did not see any improvement in performance. Generating the data for the figure took approximately 4 days on Google Colab Pro (single Tesla P100 GPU).

# B  Additional Discussion on ResNet, ResNet with BatchNorm

For Convolution Layers, the NNGP kernel is a 4-index tensor: $\mathcal{K}_{\mu\nu;ij}^l(x, x')$, where the Greek letters $(\mu, \nu)$ index the channels, whereas the Latin letters $(i, j)$ index the pixels. The infinite width limit in this case is achieved by taking the number of channels to infinity (sequentially). In this limit, most of our equations for MLP can be easily rewritten using the convolutional NNGP kernel. However, in this case, the kernel is only diagonal in channel dimension: $\mathcal{K}_{\mu\nu;ij}^l(x, x') = \mathcal{K}_{ij}^l(x, x')\delta_{\mu\nu}$. This additional structure in the kernel makes it difficult to get a closed-form solution for $\mathcal{J}^{l,l+1}$ in general.

**ResNet110 (LayerNorm)**  In Figure 4(2), the networks is critical close to $\mu = 1$, as expected from our analysis. One would naively expect the $\mu < 1$ cases also to be critical, since for MLP with ReLU and Pre-LN, $\sigma_b = 0$ is critical regardless of $\sigma_w$ and $\mu$. However, in Figure 4(2) the region away from $\mu = 1$ is in ordered phase. This is likely a result of the kernel $\mathcal{K}_{\mu\nu;ij}^l(x, x')$ not being diagonal in spatial dimensions. We emphasize that the $\mu = 1$ case stays unaffected by this, since the existence of criticality does not depend on the details of the NNGP kernel in this case. This can be readily seen from (71). We present the Numerical and training results in Figure 4.

**ResNet110 (BatchNorm)**  The operation of BatchNorm on a preactivation (pre-BN) in an MLP can be described as follows:

$$\tilde{h}_i(x) = \frac{h_i(x) - \mu_{i,B}}{\sigma_{i,B}},$$

$$\mu_{i,B} = \frac{1}{|B|} \sum_{x' \in B} h(x') \quad and \quad \sigma_{i,B} = \sqrt{\frac{1}{|B|} \sum_{x' \in B} (h(x') - \mu_{i,B})^2}, \tag{23}$$

where $B$ is the batch that $x$ belongs to and $|B|$ is batch size.

The works Yang et al. [63], He et al. [23] show that for large batch size, the effect of BatchNorm for NNGP kernel and Jacobian Norm is deterministic. We summarize the results for the pre-BN MLP setup:

$$\mathcal{K}^{l+1}(x, x') = \frac{\sigma_w^2}{N_l} \sum_{j=1}^{N_l} \mathbb{E}_\theta \left[ \phi(\tilde{h}_j^l(x)) \phi(\tilde{h}_j^l(x')) \right] + \mu^2 \mathcal{K}^l(x, x') \tag{24}$$

$$\mathcal{J}^{l,l+1} = \frac{\sigma_w^2}{K^l(x,x) - K^l(x,x')} \mathbb{E}_\theta \left[ \phi(\tilde{h}_j^l(x)) \phi(\tilde{h}_j^l(x)) \right] + \mu^2, \tag{25}$$

where $x'$ in the APJN term can be any $x' \neq x$, since for a large batch size all choices are equivalent.

From the above result, we can see that most results we have for LayerNorm can be translated to BatchNorm with an easy replacement $K^l(x,x) \to \left( \mathcal{K}^l(x,x) - \mathcal{K}^l(x,x') \right)$. As a simple example, consider a pre-BN ResNet architecture, but with all the Convolutional layers replaced with Linear (Fully Connected) layers. For such a network, we have the following result for $\mu < 1$:

$$\mathcal{J}^{l,l+1} = \frac{\pi^2(1 - \mu^2)}{(\pi - 1)^2} + \mu^2. \tag{26}$$

For $\mu = 1$, we have

$$\mathcal{J}^{l,l+1} = 1 + O(l^{-1}) \tag{27}$$

The ResNet results can then be obtained by replacing Fully Connected layers with Convolution layers, in a similar way as discussed in ResNet(LayerNorm) section. We show the numerical results for ResNet with BatchNorm in Figure 6.

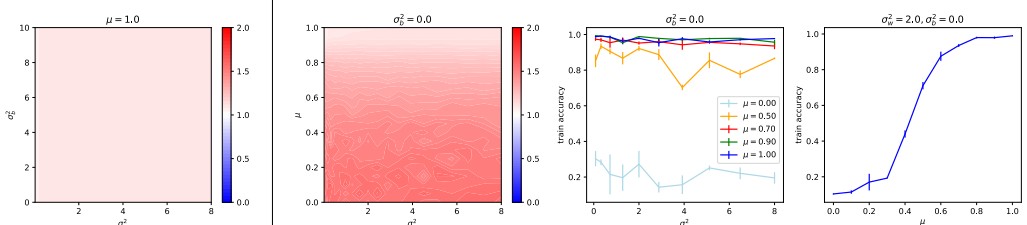

Figure 6: **ResNet110(BatchNorm)**: Left to right: (1)(2) $\mathcal{J}^{L-1,L}$ phase diagrams, with $(\sigma_w^2, \sigma_b^2)$ and $(\sigma_w^2, \mu)$. (3)(4) Training curves w.r.t $\sigma_w^2$ and $\mu$.

## C  Discussions on Transformers

As preliminary empirical results, we applied our method to Vision Transformers (ViTs). Figure 7 supports our argument in the main text that the Pre-LN architecture is generally insensitive to initialization when $\mu = 1$, where Post-LN is only stable with small $\sigma_w^2$; Figure 8 further demonstrates that the attention mechanism is highly sensitive to initialization and thus vulnerable to the gradient vanishing/diverging problem.

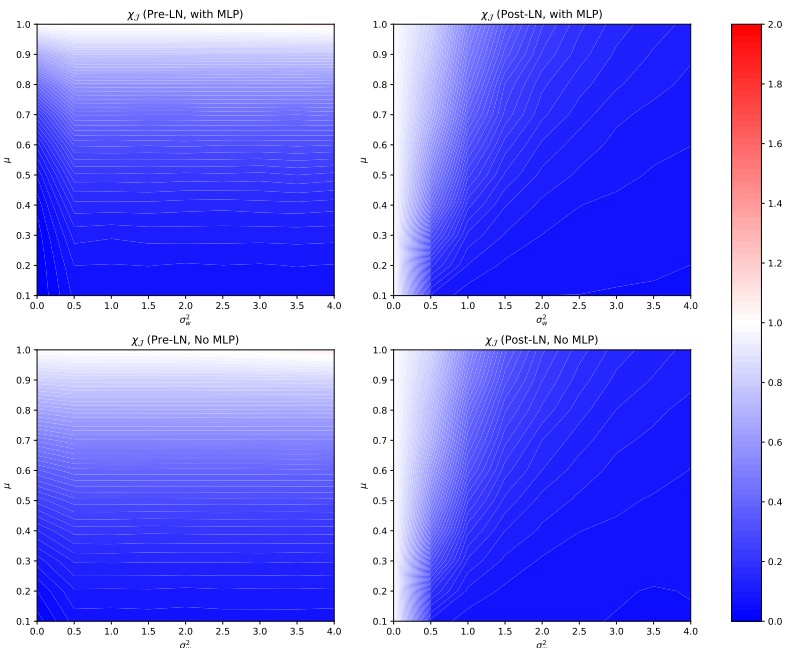

Figure 7: $\chi_{\mathcal{J}}$ Phase diagram for Vision Transformer (ViT) ($L = 24$, $n_{\text{heads}} = 8$, $d_{\text{embd}} = 256$). We present cases with pre/post-LN, with/without MLP layers. Top-left: Usual ViT – becomes everywhere-critical when $\mu = 1$; ordered otherwise. Bottom-left: Usual ViT without MLP layers. Top-right: ViT with post-LN (instead of pre-LN). Always ordered for finite $\sigma_w$. Bottom-right: ViT with post-LN, without MLP layers.

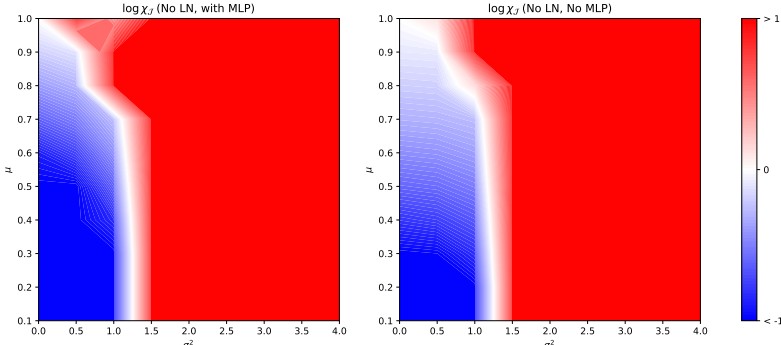

Figure 8: $\log \chi_{\mathcal{J}}$ Phase diagram for stacked Attention layers with/without MLP layers ($L = 24$, $n_{\text{heads}} = 8$, $d_{\text{embd}} = 256$). We use $\log \chi_{\mathcal{J}}$ and cut off the colorbar at $\chi_{\mathcal{J}} = 10^1$ because the $\chi_{\mathcal{J}}$ values in this case span from $\sim 50$ orders of magnitude.

## D  Technical details for Jacobians and LayerNorm

We will drop the dependence of $h_i^l(x)$ on $x$ throughout the Appendices. It should not cause any confusion since we are *always* considering a single input.

### D.1  NNGP Kernel

First, we derive the recurrence relation for the NNGP kernel Eq.(9). As mentioned in the main text, weights and biases are initialized (independently) from standard normal distribution $\mathcal{N}(0, \sigma_w^2/\text{fan\_in})$. We then have

$$\mathbb{E}_\theta[w_{ij}^l w_{mn}^l] = \frac{\sigma_w^2}{N_{l-1}} \delta_{im} \delta_{jn} \text{ and } \mathbb{E}_\theta[b_i^l b_j^l] = \sigma_b^2 \delta_{ij} \tag{28}$$

by definition.

We would like to prove theorem 2.4, as a consequence of lemma 2.2. The proof of lemma 2.2 can be found in [45].

*Proof of theorem 2.4.* One can prove this by definition with lemma 2.2.

$$
\begin{aligned}
\mathcal{K}^{l+1} &\equiv \frac{1}{N_{l+1}} \sum_{i=1}^{N_{l+1}} \mathbb{E}_\theta [h_i^{l+1} h_i^{l+1}] \\
&= \frac{1}{N_{l+1}} \sum_{i=1}^{N_{l+1}} \mathbb{E}_\theta \left[ \left( \sum_{j=1}^{N_l} w_{ij}^{l+1} \phi(h_j^l) + b_i^{l+1} \right) \left( \sum_{k=1}^{N_l} w_{ik}^{l+1} \phi(h_k^l) + b_i^{l+1} \right) \right] \\
&= \frac{1}{N_{l+1}} \sum_{i=1}^{N_{l+1}} \mathbb{E}_\theta \left[ \sum_{j=1}^{N_l} \sum_{k=1}^{N_l} w_{ij}^{l+1} w_{ik}^{l+1} \phi(h_j^l) \phi(h_k^l) + b_i^{l+1} b_i^{l+1} \right] \\
&= \frac{1}{N_{l+1}} \sum_{i=1}^{N_{l+1}} \mathbb{E}_\theta \left[ \frac{\sigma_w^2}{N_l} \sum_{j=1}^{N_l} \phi(h_j^l) \phi(h_j^l) + \sigma_b^2 \right] \\
&= \frac{\sigma_w^2}{N_l} \sum_{j=1}^{N_l} \mathbb{E}_\theta \left[ \phi(h_j^l) \phi(h_j^l) \right] + \sigma_b^2 .
\end{aligned}
\tag{29}
$$

$\square$

## D.2 Jacobians

Next, we prove theorem 2.5 in the main text.

*Proof of theorem 2.5.* We start from the definition of the averaged partial Jacobian norm (APJN) ($l > l_0$)

$$
\begin{aligned}
\mathcal{J}^{l_0,l+1} &\equiv \frac{1}{N_{l+1}} \mathbb{E}_\theta \left[ \sum_{i=1}^{N_{l+1}} \sum_{j=1}^{N_{l_0}} \frac{\partial h_i^{l+1}}{\partial h_j^{l_0}} \frac{\partial h_i^{l+1}}{\partial h_j^{l_0}} \right] \\
&= \frac{1}{N_{l+1}} \mathbb{E}_\theta \left[ \sum_{i=1}^{N_{l+1}} \sum_{j=1}^{N_{l_0}} \left( \sum_{k=1}^{N_l} \frac{\partial h_i^{l+1}}{\partial h_k^l} \frac{\partial h_k^l}{\partial h_j^{l_0}} \right) \left( \sum_{m=1}^{N_l} \frac{\partial h_i^{l+1}}{\partial h_m^l} \frac{\partial h_m^l}{\partial h_j^{l_0}} \right) \right] \\
&= \frac{1}{N_{l+1}} \mathbb{E}_\theta \left[ \sum_{i=1}^{N_{l+1}} \sum_{j=1}^{N_{l_0}} \sum_{k,m=1}^{N_l} \left( w_{ik}^{l+1} \phi'(h_k^l) \right) \left( w_{im}^{l+1} \phi'(h_m^l) \right) \left( \frac{\partial h_k^l}{\partial h_j^{l_0}} \frac{\partial h_m^l}{\partial h_j^{l_0}} \right) \right] \\
&= \frac{1}{N_{l+1}} \mathbb{E}_\theta \left[ \sum_{i=1}^{N_{l+1}} \sum_{j=1}^{N_{l_0}} \sum_{k,m=1}^{N_l} w_{ik}^{l+1} w_{im}^{l+1} \phi'(h_k^l) \phi'(h_m^l) \frac{\partial h_k^l}{\partial h_j^{l_0}} \frac{\partial h_m^l}{\partial h_j^{l_0}} \right] \\
&= \frac{1}{N_{l+1}} \sum_{i=1}^{N_{l+1}} \sum_{j=1}^{N_{l_0}} \sum_{k=1}^{N_l} \frac{\sigma_w^2}{N_l} \mathbb{E}_\theta \left[ \phi'(h_k^l) \phi'(h_k^l) \frac{\partial h_k^l}{\partial h_j^{l_0}} \frac{\partial h_k^l}{\partial h_j^{l_0}} \right] \\
&= \frac{\sigma_w^2}{N_l} \sum_{k=1}^{N_l} \mathbb{E}_\theta \left[ \phi'(h_k^l) \phi'(h_k^l) \left( \sum_{j=1}^{N_{l_0}} \frac{\partial h_k^l}{\partial h_j^{l_0}} \frac{\partial h_k^l}{\partial h_j^{l_0}} \right) \right] .
\end{aligned}
\tag{30}
$$

where we used the chain rule and took the expectation value over $w^{l+1}$. Next, we take the chain rule again:

$$
\begin{aligned}
\mathcal{J}^{l_0,l+1} &= \frac{\sigma_w^2}{N_l} \sum_{k=1}^{N_l} \mathbb{E}_\theta \left[ \phi'(h_k^l)\phi'(h_k^l) \left( \sum_{j=1}^{N_{l_0}} \sum_{m,n=1}^{N_{l-1}} w_{km}^l w_{kn}^l \phi'(h_m^{l-1})\phi'(h_n^{l-1}) \frac{\partial h_m^{l-1}}{\partial h_j^{l_0}} \frac{\partial h_n^{l-1}}{\partial h_j^{l_0}} \right) \right] \\
&= \frac{\sigma_w^4}{N_l N_{l-1}} \sum_{k=1}^{N_l} \mathbb{E}_\theta \left[ \phi'(h_k^l)\phi'(h_k^l) \left( \sum_{j=1}^{N_{l_0}} \sum_{m=1}^{N_{l-1}} \phi'(h_m^{l-1})\phi'(h_m^{l-1}) \frac{\partial h_m^{l-1}}{\partial h_j^{l_0}} \frac{\partial h_m^{l-1}}{\partial h_j^{l_0}} \right) + O(1/N_{l-1}) \right] \\
&= \frac{\sigma_w^4}{N_l N_{l-1}} \sum_{k=1}^{N_l} \mathbb{E}_\theta \left[ \phi'(h_k^l)\phi'(h_k^l) \right] \mathbb{E}_\theta \left[ \left( \sum_{j=1}^{N_{l_0}} \sum_{m=1}^{N_{l-1}} \phi'(h_m^{l-1})\phi'(h_m^{l-1}) \frac{\partial h_m^{l-1}}{\partial h_j^{l_0}} \frac{\partial h_m^{l-1}}{\partial h_j^{l_0}} \right) \right] \\
&= \frac{\sigma_w^2}{N_l} \sum_{k=1}^{N_l} \mathbb{E}_\theta \left[ \phi'(h_k^l)\phi'(h_k^l) \right] \cdot \frac{\sigma_w^2}{N_{l-1}} \mathbb{E}_\theta \left[ \left( \sum_{j=1}^{N_{l_0}} \sum_{m=1}^{N_{l-1}} \phi'(h_m^{l-1})\phi'(h_m^{l-1}) \frac{\partial h_m^{l-1}}{\partial h_j^{l_0}} \frac{\partial h_m^{l-1}}{\partial h_j^{l_0}} \right) \right] \\
&= \chi_{\mathcal{J}}^l \mathcal{J}^{l_0,l} ,
\end{aligned}
\tag{31}
$$

where we integrate (by parts) over $w^l$ to get the second line. We take $N_{l-1} \to \infty$ and then $N_l \to \infty$ to get the third line. We rearrange terms and use the Eq.(11) to get the fourth and fifth lines. Notice that to get the third line we used the fact in the infinite width limit, the distribution of $h_i^l$ is independent of $h_i^{l-1}$. Thus we proved

$$
\mathcal{J}^{l_0,l+1} = \chi_{\mathcal{J}}^l \mathcal{J}^{l_0,l} .
\tag{32}
$$

$\square$

The critical line is defined by requiring $\chi_{\mathcal{J}}^\star = 1$, where critical points are reached by further requiring $\chi_{\mathcal{K}}^\star = 1$.

As we mentioned in the main text, $l_0 = 0$ is subtle since the input dimension is fixed $N_0$, which can not be assumed to be infinity. Even though for a dataset like MNIST, usually $N_0$ is not significantly smaller than width $N_l$. We show how to take finite $O(N_0^{-1})$ correction into account by using one example.

**Lemma D.1.** *Consider a one hidden layer network with a finite input dimension $N_0$. In the infinite width limit, the APJN is still deterministic and the first step of the recurrence relation is modified to:*

$$
\mathcal{J}^{0,2} = \left( \chi_{\mathcal{J}}^1 + \frac{2\sigma_w^2}{N_0} \chi_\Delta^1 \sum_k^{N_0} \frac{1}{N_0} h_k^0 h_k^0 \right) \mathcal{J}^{0,1} ,
\tag{33}
$$

*where $\mathcal{J}^{0,1} = \sigma_w^2$.*

*Proof.*

$$
\begin{aligned}
\mathcal{J}^{0,2} =& \frac{1}{N_2} \mathbb{E}_\theta \left[ \sum_{i=1}^{N_2} \sum_{j=1}^{N_0} \frac{\partial h_i^2}{\partial h_j^0} \frac{\partial h_i^2}{\partial h_j^0} \right] \\
=& \frac{1}{N_2} \mathbb{E}_\theta \left[ \sum_{i=1}^{N_2} \sum_{j=1}^{N_0} \sum_{k,m=1}^{N_1} w_{ik}^2 w_{im}^2 \phi'(h_k^1) \phi'(h_m^1) \frac{\partial h_k^1}{\partial h_j^0} \frac{\partial h_m^1}{\partial h_j^0} \right] \\
=& \frac{1}{N_2} \sum_{i=1}^{N_2} \sum_{j=1}^{N_0} \sum_{k,m=1}^{N_1} \mathbb{E}_\theta [ w_{ik}^2 w_{im}^2 w_{kj}^1 w_{mj}^1 \phi'(h_k^1) \phi'(h_m^1) ] \\
=& \sum_{j=1}^{N_0} \sum_{k=1}^{N_1} \frac{\sigma_w^2}{N_1} \mathbb{E}_\theta [ w_{kj}^1 w_{kj}^1 \phi'(h_k^1) \phi'(h_k^1) ] \\
=& \sigma_w^2 \left( \chi_{\mathcal{J}}^1 + \frac{2\sigma_w^2}{N_0} \chi_\Delta^1 \sum_k^{N_0} \frac{1}{N_0} h_k^0 h_k^0 \right) \\
=& \left( \chi_{\mathcal{J}}^1 + \frac{2\sigma_w^2}{N_0} \chi_\Delta^1 \sum_k^{N_0} \frac{1}{N_0} h_k^0 h_k^0 \right) \mathcal{J}^{0,1} ,
\end{aligned}
\tag{34}
$$

where to get the result we used integrate by parts, then explicitly integrated over $w_{ij}^1$. We have introduced a coefficient of finite width corrections, $\chi_\Delta^l$, defined as follows. $\qquad\square$

**Definition D.2** (Coefficient of Finite Width Corrections)**.**

$$
\chi_\Delta^l = \frac{\sigma_w^2}{N_l} \sum_{i=1}^{N_l} \mathbb{E}_\theta [ \phi''(h_i^l) \phi''(h_i^l) + \phi'''(h_i^l) \phi'(h_i^l) ] .
\tag{35}
$$

*Remark* D.3. Notice that the correction to $\mathcal{J}^{0,2}$ is order $O(N_0^{-1})$. If one calculates the recurrence relation for deeper layers, the correction to $\mathcal{J}^{0,l}$ will be $O(\sum_{l'=0}^{l} N_{l'}^{-1})$, which means the contribution from hidden layers can be ignored in infinite width limit.

The $\mathcal{J}^{0,2}$ example justifies the factorization of the integral when we go from the last line of Eq.(30) to Eq.(32).

Finally, the full Jacobian in infinite width limit can be written as

**Lemma D.4** (APJN with $l_0 = 0$)**.** *The APJN (with $l_0 = 0$) of a given network can be written as*

$$
\mathcal{J}^{0,l} = \sigma_w^2 \left( \chi_{\mathcal{J}}^1 + \frac{2\sigma_w^2}{N_0} \chi_\Delta^1 \sum_k^{N_0} \frac{1}{N_0} h_k^0 h_k^0 \right) \prod_{l'=2}^{l-1} \chi_{\mathcal{J}}^{l'} .
\tag{36}
$$

*Note that APJN with $l_0 > 0$ does not receive the $O(N_0^{-1})$ correction.*

### D.3 APJN and gradients

As mentioned in the main text, APJN is an important tool in studying the exploding and vanishing gradients problem. Its utility stems from the fact that it is a dominant factor in the norm of the gradients. This can be readily by looking at the (squared) $L_2$ norm of the gradient of any flattened

parameter matrix $\theta^l$, at initialization. In the infinite width limit, one gets

$$\|\nabla_{\theta^l}\mathcal{L}\|_2^2 = \left(\sum_{all} \frac{\partial\mathcal{L}}{\partial h_i^L}\frac{\partial h_i^L}{\partial h_j^{L-1}}\cdots\frac{\partial h_k^{l+1}}{\partial h_m^l}\frac{\partial h_m^l}{\partial \theta_n^l}\right)^2$$

$$= \sum_{all}\left(\frac{\partial\mathcal{L}}{\partial h_i^L}\frac{\partial\mathcal{L}}{\partial h_{i'}^L}\right)\left(\frac{\partial h_i^L}{\partial h_j^{L-1}}\frac{\partial h_{i'}^L}{\partial h_{j'}^{L-1}}\right)\cdots\left(\frac{\partial h_k^{l+1}}{\partial h_m^l}\frac{\partial h_{k'}^{l+1}}{\partial h_{m'}^l}\right)\left(\frac{\partial h_m^l}{\partial \theta_n^l}\frac{\partial h_{m'}^l}{\partial \theta_n^l}\right)$$

$$= \sum_{all}\left(\frac{\partial\mathcal{L}}{\partial h_i^L}\frac{\partial\mathcal{L}}{\partial h_{i'}^L}\right)\left(\frac{\partial h_i^L}{\partial h_j^{L-1}}\frac{\partial h_{i'}^L}{\partial h_{j'}^{L-1}}\right)\cdots\left(\frac{\partial h_k^{l+1}}{\partial h_m^l}\frac{\partial h_{k'}^{l+1}}{\partial h_{m'}^l}\right)\delta_{mm'}\left\|\frac{\partial h^l}{\partial \theta^l}\right\|_F^2$$

$$= \sum_{all}\left(\frac{\partial\mathcal{L}}{\partial h_i^L}\frac{\partial\mathcal{L}}{\partial h_{i'}^L}\right)\left(\frac{\partial h_i^L}{\partial h_j^{L-1}}\frac{\partial h_{i'}^L}{\partial h_{j'}^{L-1}}\right)\cdots\delta_{kk'}\mathcal{J}^{l,l+1}\left\|\frac{\partial h^l}{\partial \theta^l}\right\|_F^2$$

$$= \sum_{i,i',j,j'}\left(\frac{\partial\mathcal{L}}{\partial h_i^L}\frac{\partial\mathcal{L}}{\partial h_{i'}^L}\right)\left(\frac{\partial h_i^L}{\partial h_j^{L-1}}\frac{\partial h_{i'}^L}{\partial h_{j'}^{L-1}}\right)\delta_{jj'}\cdots\mathcal{J}^{l,l+1}\left\|\frac{\partial h^l}{\partial \theta^l}\right\|_F^2$$

$$= \sum_{i,i'}\left(\frac{\partial\mathcal{L}}{\partial h_i^L}\frac{\partial\mathcal{L}}{\partial h_{i'}^L}\right)\delta_{ii'}\mathcal{J}^{L-1,L}\cdots\mathcal{J}^{l,l+1}\left\|\frac{\partial h^l}{\partial \theta^l}\right\|_F^2$$

$$= \left\|\frac{\partial\mathcal{L}}{\partial h^L}\right\|_2^2\mathcal{J}^{L-1,L}\cdots\mathcal{J}^{l,l+1}\left\|\frac{\partial h^l}{\partial \theta^l}\right\|_F^2$$

$$= \left\|\frac{\partial\mathcal{L}}{\partial h^L}\right\|_2^2\mathcal{J}^{l,L}\left\|\frac{\partial h^l}{\partial \theta^l}\right\|_F^2 , \tag{37}$$

where $\|\cdot\|_2$ denotes the $L_2$ norm and $\|\cdot\|_F$ denotes the Frobenius norm.

## D.4 LayerNorm on Pre-activations

**Definition D.5** (Layer Normalization).

$$\tilde{h}_i^l = \frac{h_i^l - \mathbb{E}[h^l]}{\sqrt{\mathbb{E}[(h^l)^2] - \mathbb{E}[h^l]^2}}\gamma_i^l + \beta_i^l , \tag{38}$$

where $\gamma_i^l$ and $\beta_i^l$ are learnable parameters.

*Remark* D.6. With only LayerNorm, the (1) is simplified to

$$h_i^{l+1} = \sum_{j=1}^{N_l} w_{ij}^{l+1}\phi(\tilde{h}_j^l) + b_i^{l+1} . \tag{39}$$

*Remark* D.7. In the limit of infinite width, using the law of large numbers, the average over neurons $\mathbb{E}[\cdots]$ can be replaced by the average of parameter-initializations $\mathbb{E}_\theta[\cdots]$. Additionally, in this limit, the preactivations are i.i.d. Gaussian distributed : $h^l \sim \mathcal{N}(0, \mathcal{K}^l)$.

$$\mathbb{E}\left[h^l\right] = \mathbb{E}_\theta\left[h^l\right] = 0 , \tag{40}$$

$$\mathbb{E}\left[(h^l)^2\right] = \mathbb{E}_\theta\left[(h^l)^2\right] = \mathcal{K}^l . \tag{41}$$

The normalized preactivation then simplifies to the form of Eq.(19).

*Remark* D.8. At initialization, the parameters $\gamma_i^l$ and $\beta_i^l$ take the values 1 and 0, respectively. This leads to the form in equation (19). In infinite width limit, it has the following form

$$\tilde{h}_i^l = \frac{h_i^l - \mathbb{E}_\theta[h^l]}{\sqrt{\mathbb{E}_\theta[(h^l)^2] - \mathbb{E}_\theta[h^l]^2}} . \tag{42}$$

**Lemma D.9.** *With LayerNorm on preactivations, the gaussian average is modified to*

$$\mathbb{E}_\theta\left[O(\tilde{h}_i^l)\right] = \frac{1}{\sqrt{2\pi}}\int d\tilde{h}_i^l\, O(\tilde{h}_i^l)\, e^{-\frac{(\tilde{h}_i^l)^2}{2}} . \tag{43}$$

*Proof.* By definition $\tilde{h}_i^l$ is sampled from a standard normal distribution $\mathcal{N}(0, 1)$, then use lemma 2.2 to get the final form. □

**Theorem D.10.** *In the infinite width limit the recurrence relation for the NNGP kernel with Layer-Norm on preactivations is*

$$\mathcal{K}^{l+1} = \frac{\sigma_w^2}{N_l} \sum_{j=1}^{N_l} \mathbb{E}_\theta \left[ \phi(\tilde{h}_j^l)\phi(\tilde{h}_j^l) \right] + \sigma_b^2 \,. \tag{44}$$

*Proof.*

$$
\begin{aligned}
\mathcal{K}^{l+1} =& \frac{1}{N_{l+1}} \sum_{i=1}^{N_{l+1}} \mathbb{E}_\theta \left[ h_i^{l+1} h_i^{l+1} \right] \\
=& \frac{1}{N_{l+1}} \sum_{i=1}^{N_{l+1}} \mathbb{E}_\theta \left[ \left( \sum_{j=1}^{N_l} w_{ij}^{l+1}\phi(\tilde{h}_j^l) + b_i^{l+1} \right) \left( \sum_{k=1}^{N_l} w_{ik}^{l+1}\phi(\tilde{h}_k^l) + b_i^{l+1} \right) \right] \\
=& \frac{\sigma_w^2}{N_l} \sum_{j=1}^{N_l} \mathbb{E}_\theta \left[ \phi(\tilde{h}_j^l)\phi(\tilde{h}_j^l) \right] + \sigma_b^2 \,.
\end{aligned}
\tag{45}
$$

□

**Theorem D.11.** *In the infinite width limit the recurrence relation for partial Jacobian with LayerNorm on preactivations is*

$$\mathcal{J}^{l_0,l+1} = \chi_{\mathcal{J}}^l \mathcal{J}^{l_0,l} \,, \tag{46}$$

*where $\chi_{\mathcal{J}}^l = \frac{\sigma_w^2}{N_l \mathcal{K}^l} \sum_{i=1}^{N_l} \mathbb{E}_\theta \left[ \phi'(\tilde{h}_i^l)^2 \right]$.*

*Proof.*

$$
\begin{aligned}
\mathcal{J}^{l_0,l+1} =& \frac{1}{N_{l+1}} \mathbb{E}_\theta \left[ \sum_{i=1}^{N_{l+1}} \sum_{j=1}^{N_{l_0}} \frac{\partial h_i^{l+1}}{\partial h_j^{l_0}} \frac{\partial h_i^{l+1}}{\partial h_j^{l_0}} \right] \\
=& \frac{1}{N_{l+1}} \mathbb{E}_\theta \left[ \sum_{i=1}^{N_{l+1}} \sum_{j=1}^{N_{l_0}} \left( \sum_{k=1}^{N_l} \frac{\partial h_i^{l+1}}{\partial \tilde{h}_k^l} \frac{\partial \tilde{h}_k^l}{\partial h_k^l} \frac{\partial h_k^l}{\partial h_j^{l_0}} \right) \left( \sum_{m=1}^{N_l} \frac{\partial h_i^{l+1}}{\partial \tilde{h}_m^l} \frac{\partial \tilde{h}_m^l}{\partial h_m^l} \frac{\partial h_m^l}{\partial h_j^{l_0}} \right) \right] \\
=& \frac{1}{N_{l+1}} \mathbb{E}_\theta \left[ \sum_{i=1}^{N_{l+1}} \sum_{j=1}^{N_{l_0}} \sum_{k,m=1}^{N_l} \left( w_{ik}^{l+1}\phi'(\tilde{h}_k^l)\frac{1}{\sqrt{\mathcal{K}^l}} \right) \left( w_{im}^{l+1}\phi'(\tilde{h}_m^l)\frac{1}{\sqrt{\mathcal{K}^l}} \right) \left( \frac{\partial h_k^l}{\partial h_j^{l_0}} \frac{\partial h_m^l}{\partial h_j^{l_0}} \right) \right] \\
=& \frac{\sigma_w^2}{N_l \mathcal{K}^l} \sum_{k=1}^{N_l} \mathbb{E}_\theta \left[ \phi'(\tilde{h}_k^l)\phi'(\tilde{h}_k^l) \left( \sum_{j=1}^{N_{l_0}} \frac{\partial h_k^l}{\partial h_j^{l_0}} \frac{\partial h_k^l}{\partial h_j^{l_0}} \right) \right] \\
=& \frac{\sigma_w^2}{N_l \mathcal{K}^l} \sum_{k=1}^{N_l} \mathbb{E}_\theta \left[ \phi'(\tilde{h}_k^l)\phi'(\tilde{h}_k^l) \right] \mathcal{J}^{l_0,l} \\
=& \chi_{\mathcal{J}}^l \mathcal{J}^{l_0,l} \,,
\end{aligned}
\tag{47}
$$

□

## D.5 LayerNorm on activations

The general definition of LayerNorm on activations is given as follows.

**Definition D.12** (LayerNorm on Activations)**.**

$$\widetilde{\phi(h_i^l)} = \frac{\phi(h_i^l) - \mathbb{E}[\phi(h^l)]}{\sqrt{\mathbb{E}[\phi(h^l)^2] - \mathbb{E}[\phi(h^l)]^2}} \gamma_i^l + \beta_i^l \,. \tag{48}$$

*Remark* D.13. The recurrence relation for preactivations (Eq.(1)) gets modified to

$$h_i^{l+1} = \sum_{j=1}^{N_l} w_{ij}^{l+1} \widetilde{\phi(h_j^l)} + b_i^{l+1} \,. \tag{49}$$

*Remark* D.14. At initialization, the parameters $\gamma_i^l$ and $\beta_i^l$ take the values 1 and 0, respectively. This leads to the form

$$
\begin{aligned}
\widetilde{\phi(h_i^l)} &= \frac{\phi(h_i^l) - \mathbb{E}[\phi(h^l)]}{\sqrt{\mathbb{E}[\phi(h^l)^2] - \mathbb{E}[\phi(h^l)]^2}} \\
&= \frac{\phi(h_i^l) - \mathbb{E}_\theta\left[\phi(h^l)\right]}{\sqrt{\mathbb{E}_\theta\left[\phi(h^l)^2\right] - \mathbb{E}_\theta\left[\phi(h^l)\right]^2}} \,,
\end{aligned}
\tag{50}
$$

where the first line follows from the fact that at initialization, the parameters $\gamma_i^l$ and $\beta_i^l$ take the values 1 and 0 respectively. In the second line, we have invoked the infinite width limit.

*Remark* D.15. Evaluating the Gaussian average in this case is similar to the cases in the previous section. The only difference is that the averages are taking over the distribution $h^{l-1} \sim \mathcal{N}(0, \mathcal{K}^{l-1} = \sigma_w^2 + \sigma_b^2)$. Again this can be summarized as

$$\mathbb{E}_\theta\left[O(h_i^l)\right] = \frac{1}{\sqrt{2\pi(\sigma_w^2 + \sigma_b^2)}} \int dh_i^l \, O(h_i^l) \, e^{-\frac{(h_i^l)^2}{2(\sigma_w^2 + \sigma_b^2)}} \,. \tag{51}$$

Next, we calculate the modifications to the recurrence relations for the NNGP kernel and Jacobians.

**Theorem D.16.** *In the infinite width limit the recurrence relation for the NNGP kernel with Layer-Norm on activations is*

$$\mathcal{K}^{l+1} = \sigma_w^2 + \sigma_b^2 \,. \tag{52}$$

*Proof.*

$$
\begin{aligned}
\mathcal{K}^{l+1} &= \frac{1}{N_{l+1}} \sum_{i=1}^{N_{l+1}} \mathbb{E}_\theta\left[h_i^{l+1} h_i^{l+1}\right] \\
&= \frac{1}{N_{l+1}} \sum_{i=1}^{N_{l+1}} \mathbb{E}_\theta\left[\left(\sum_{j=1}^{N_l} w_{ij}^{l+1} \widetilde{\phi(h_j^l)} + b_i^{l+1}\right)\left(\sum_{k=1}^{N_l} w_{ik}^{l+1} \widetilde{\phi(h_k^l)} + b_i^{l+1}\right)\right] \\
&= \frac{\sigma_w^2}{N_l} \sum_{j=1}^{N_l} \mathbb{E}_\theta\left[\widetilde{\phi(h_j^l)}^2\right] + \sigma_b^2 \\
&= \frac{\sigma_w^2}{N_l} \sum_{j=1}^{N_l} \mathbb{E}_\theta\left[\left(\frac{\phi(h_j^l) - \mathbb{E}_\theta\left[\phi(h^l)\right]}{\sqrt{\mathbb{E}_\theta\left[\phi(h^l)^2\right] - \mathbb{E}_\theta\left[\phi(h^l)\right]^2}}\right)^2\right] + \sigma_b^2 \\
&= \frac{\sigma_w^2}{N_l} \sum_{j=1}^{N_l} \frac{\mathbb{E}_\theta\left[\left(\phi(h_j^l) - \mathbb{E}_\theta\left[\phi(h^l)\right]\right)^2\right]}{\mathbb{E}_\theta\left[\phi(h^l)^2\right] - \mathbb{E}_\theta\left[\phi(h^l)\right]^2} + \sigma_b^2 \\
&= \sigma_w^2 + \sigma_b^2 \,. 
\end{aligned}
\tag{53}
$$

$\square$

**Theorem D.17.** *In the infinite width limit the recurrence relation for partial Jacobian with LayerNorm on activations is*

$$\mathcal{J}^{l_0, l+1} = \chi_J^l \mathcal{J}^{l_0, l} \,, \tag{54}$$

*where* $\chi_J^l \equiv \sigma_w^2 \frac{\mathbb{E}_\theta\left[\phi'(h^l)^2\right]}{\mathbb{E}_\theta\left[\phi(h^l)^2\right] - \mathbb{E}_\theta\left[\phi(h^l)\right]^2}$.

*Proof.*

$$
\begin{aligned}
\mathcal{J}^{l_0,l+1} &= \frac{1}{N_{l+1}} \mathbb{E}_\theta \left[ \sum_{i=1}^{N_{l+1}} \sum_{j=1}^{N_{l_0}} \frac{\partial h_i^{l+1}}{\partial h_j^{l_0}} \frac{\partial h_i^{l+1}}{\partial h_j^{l_0}} \right] \\
&= \frac{1}{N_{l+1}} \mathbb{E}_\theta \left[ \sum_{i=1}^{N_{l+1}} \sum_{j=1}^{N_{l_0}} \left( \sum_{k=1}^{N_l} \frac{\partial h_i^{l+1}}{\partial h_k^l} \frac{\partial h_k^l}{\partial h_j^{l_0}} \right) \left( \sum_{m=1}^{N_l} \frac{\partial h_i^{l+1}}{\partial h_m^l} \frac{\partial h_m^l}{\partial h_j^{l_0}} \right) \right] \\
&= \frac{1}{N_{l+1}} \mathbb{E}_\theta \left[ \sum_{i=1}^{N_{l+1}} \sum_{j=1}^{N_{l_0}} \sum_{k,m=1}^{N_l} \left( w_{ik}^{l+1} \widetilde{\phi'(h_k^l)} \right) \left( w_{im}^{l+1} \widetilde{\phi'(h_m^l)} \right) \left( \frac{\partial h_k^l}{\partial h_j^{l_0}} \frac{\partial h_m^l}{\partial h_j^{l_0}} \right) \right] \\
&= \frac{\sigma_w^2}{N_l} \sum_{k=1}^{N_l} \sum_{j=1}^{N_{l_0}} \mathbb{E}_\theta \left[ \widetilde{\phi'(h_k^l)} \widetilde{\phi'(h_k^l)} \left( \sum_{j=1}^{N_{l_0}} \frac{\partial h_k^l}{\partial h_j^{l_0}} \frac{\partial h_k^l}{\partial h_j^{l_0}} \right) \right] \\
&= \frac{\sigma_w^2}{N_l} \sum_{k=1}^{N_l} \mathbb{E}_\theta \left[ \widetilde{\phi'(h_k^l)}^2 \right] \mathcal{J}^{l_0,l} \\
&= \sigma_w^2 \frac{\mathbb{E}_\theta \left[ \phi'(h^l)^2 \right]}{\mathbb{E}_\theta \left[ \phi(h^l)^2 \right] - \mathbb{E}_\theta \left[ \phi(h^l) \right]^2} \mathcal{J}^{l_0,l} \\
&= \chi_J^l \mathcal{J}^{l_0,l},
\end{aligned}
\tag{55}
$$

$\square$

## E Residual Connections

**Definition E.1.** We define residual connections by the modified the recurrence relation for preactivations (Eq.(1))

$$
h_i^{l+1} = \sum_{j=1}^{N_l} w_{ij}^{l+1} \phi(h_j^l) + b_i^{l+1} + \mu h_i^l,
\tag{56}
$$

where the parameter $\mu$ controls the strength of the residual connection.

*Remark* E.2. Note that this definition requires $N_{l+1} = N_l$. We ensure this by only adding residual connections to the hidden layers, which are of the same width. More generally, one can introduce a tensor parameter $\mu_{ij}$.

*Remark* E.3. In general, the parameter $\mu$ could be layer-dependent ($\mu^l$). But we suppress this dependence here since we are discussing self-similar networks.

**Theorem E.4.** *In the infinite width limit, the recurrence relation for the NNGP kernel with residual connections is changed by an additional term controlled by $\mu$*

$$
\mathcal{K}^{l+1} = \frac{\sigma_w^2}{N_l} \sum_{j=1}^{N_l} \mathbb{E}_\theta \left[ \phi(h_j^l) \phi(h_j^l) \right] + \sigma_b^2 + \mu^2 \mathcal{K}^l.
\tag{57}
$$

*Proof.*

$$\mathcal{K}^{l+1} = \frac{1}{N_{l+1}} \sum_{i=1}^{N_{l+1}} \mathbb{E}_\theta \left[ h_i^{l+1} h_i^{l+1} \right]$$

$$= \frac{1}{N_{l+1}} \sum_{i=1}^{N_{l+1}} \mathbb{E}_\theta \left[ \left( \sum_{j=1}^{N_l} w_{ij}^{l+1} \phi(h_j^l) + b_i^{l+1} + \mu h_i^l \right) \left( \sum_{k=1}^{N_l} w_{ik}^{l+1} \phi(h_k^l) + b_i^{l+1} + \mu h_i^l \right) \right]$$

$$= \frac{1}{N_{l+1}} \sum_{i=1}^{N_{l+1}} \mathbb{E}_\theta \left[ \sum_{j=1}^{N_l} \sum_{k=1}^{N_l} w_{ij}^{l+1} w_{ik}^{l+1} \phi(h_j^l) \phi(h_k^l) + b_i^{l+1} b_i^{l+1} + \mu^2 h_i^l h_i^l \right]$$

$$= \frac{1}{N_{l+1}} \sum_{i=1}^{N_{l+1}} \mathbb{E}_\theta \left[ \frac{\sigma_w^2}{N_l} \sum_{j=1}^{N_l} \phi(h_j^l) \phi(h_j^l) + \sigma_b^2 \right] + \mu^2 \frac{1}{N_{l+1}} \sum_{i=1}^{N_{l+1}} \mathbb{E}_\theta \left[ h_i^l h_i^l \right]$$

$$= \frac{\sigma_w^2}{N_l} \sum_{j=1}^{N_l} \mathbb{E}_\theta \left[ \phi(h_j^l) \phi(h_j^l) \right] + \sigma_b^2 + \mu^2 \mathcal{K}^l \,, \tag{58}$$

where we used the fact $N_{l+1} = N_l$ to get the last line. $\qquad\square$

**Theorem E.5.** *In the infinite width limit, the recurrence relation for partial Jacobians with residual connections has a simple multiplicative form*

$$\mathcal{J}^{l_0,l+1} = \chi_\mathcal{J}^l \mathcal{J}^{l_0,l} \,, \tag{59}$$

*where the recurrence coefficient is shifted to* $\chi_\mathcal{J}^l = \sigma_w^2 \mathbb{E}_\theta \left[ \phi'(h_k^l) \phi'(h_k^l) \right] + \mu^2$.

*Proof.*

$$\mathcal{J}^{l_0,l+1} \equiv \frac{1}{N_{l+1}} \mathbb{E}_\theta \left[ \sum_{i=1}^{N_{l+1}} \sum_{j=1}^{N_{l_0}} \frac{\partial h_i^{l+1}}{\partial h_j^{l_0}} \frac{\partial h_i^{l+1}}{\partial h_j^{l_0}} \right]$$

$$= \frac{1}{N_{l+1}} \mathbb{E}_\theta \left[ \sum_{i=1}^{N_{l+1}} \sum_{j=1}^{N_{l_0}} \left( \sum_{k=1}^{N_l} \frac{\partial h_i^{l+1}}{\partial h_k^l} \frac{\partial h_k^l}{\partial h_j^{l_0}} \right) \left( \sum_{m=1}^{N_l} \frac{\partial h_i^{l+1}}{\partial h_m^l} \frac{\partial h_m^l}{\partial h_j^{l_0}} \right) \right]$$

$$= \frac{1}{N_{l+1}} \mathbb{E}_\theta \left[ \sum_{i=1}^{N_{l+1}} \sum_{j=1}^{N_{l_0}} \sum_{k,m=1}^{N_l} \left( w_{ik}^{l+1} \phi'(h_k^l) + \mu \delta_{ik} \right) \left( w_{im}^{l+1} \phi'(h_m^l) + \mu \delta_{im} \right) \left( \frac{\partial h_k^l}{\partial h_j^{l_0}} \frac{\partial h_m^l}{\partial h_j^{l_0}} \right) \right]$$

$$= \frac{1}{N_{l+1}} \mathbb{E}_\theta \left[ \sum_{i=1}^{N_{l+1}} \sum_{j=1}^{N_{l_0}} \sum_{k,m=1}^{N_l} \left( w_{ik}^{l+1} w_{im}^{l+1} \phi'(h_k^l) \phi'(h_m^l) + \mu^2 \delta_{ik} \delta_{im} \right) \frac{\partial h_k^l}{\partial h_j^{l_0}} \frac{\partial h_m^l}{\partial h_j^{l_0}} \right]$$

$$= \frac{\sigma_w^2}{N_l} \sum_{k=1}^{N_l} \mathbb{E}_\theta \left[ \phi'(h_k^l) \phi'(h_k^l) \left( \sum_{j=1}^{N_{l_0}} \frac{\partial h_k^l}{\partial h_j^{l_0}} \frac{\partial h_k^l}{\partial h_j^{l_0}} \right) \right] + \frac{1}{N_l} \sum_{k=1}^{N_l} \mathbb{E}_\theta \left[ \mu^2 \left( \sum_{j=1}^{N_{l_0}} \frac{\partial h_k^l}{\partial h_j^{l_0}} \frac{\partial h_k^l}{\partial h_j^{l_0}} \right) \right]$$

$$= \left( \sigma_w^2 \mathbb{E}_\theta \left[ \phi'(h_k^l) \phi'(h_k^l) \right] + \mu^2 \right) \mathbb{E}_\theta \left[ \frac{1}{N_l} \sum_{k=1}^{N_l} \sum_{j=1}^{N_{l_0}} \frac{\partial h_k^l}{\partial h_j^{l_0}} \frac{\partial h_k^l}{\partial h_j^{l_0}} \right]$$

$$= \left( \sigma_w^2 \mathbb{E}_\theta \left[ \phi'(h_k^l) \phi'(h_k^l) \right] + \mu^2 \right) \mathcal{J}^{l_0,l}$$

$$\mathcal{J}^{l_0,l+1} = \chi_\mathcal{J}^l \mathcal{J}^{l_0,l} \,. \tag{60}$$

$\qquad\square$

# F    Residual Connections with LayerNorm on Preactivations (Pre-LN)

We recall the recurrence relation (1):

$$h_i^{l+1} = \sum_{j=1}^{N_l} w_{ij}^{l+1} \phi(\tilde{h}_j^l) + b_i^{l+1} + \mu h_i^l \,. \tag{61}$$

**Theorem F.1.** *In the infinite width limit, the recurrence relation for the NNGP kernel is then modified to*

$$\mathcal{K}^{l+1} = \frac{\sigma_w^2}{N_l} \sum_{j=1}^{N_l} \mathbb{E}_\theta \left[ \phi(\tilde{h}_j^l)\phi(\tilde{h}_j^l) \right] + \sigma_b^2 + \mu^2 \mathcal{K}^l \,. \tag{62}$$

*Proof.*

$$
\begin{aligned}
\mathcal{K}^{l+1} &= \frac{1}{N_{l+1}} \sum_{i=1}^{N_{l+1}} \mathbb{E}_\theta \left[ h_i^{l+1} h_i^{l+1} \right] \\
&= \frac{1}{N_{l+1}} \sum_{i=1}^{N_{l+1}} \mathbb{E}_\theta \left[ \left( \sum_{j=1}^{N_l} w_{ij}^{l+1} \phi(\tilde{h}_j^l) + b_i^{l+1} + \mu h_i^l \right) \left( \sum_{k=1}^{N_l} w_{ik}^{l+1} \phi(\tilde{h}_k^l) + b_i^{l+1} + \mu h_i^l \right) \right] \\
&= \frac{\sigma_w^2}{N_l} \sum_{j=1}^{N_l} \mathbb{E}_\theta \left[ \phi(\tilde{h}_j^l)\phi(\tilde{h}_j^l) \right] + \sigma_b^2 + \mu^2 \mathcal{K}^l \,.
\end{aligned}
\tag{63}
$$

$\square$

*Remark* F.2.  For $\mu < 1$, the recursion relation has a fixed point

$$\mathcal{K}^\star = \frac{\sigma_w^2}{N_{l^\star}(1-\mu^2)} \sum_{j=1}^{N_{l^\star}} \mathbb{E}_\theta \left[ \phi(\tilde{h}_j^{l^\star})\phi(\tilde{h}_j^{l^\star}) \right] + \frac{\sigma_b^2}{1-\mu^2} \,. \tag{64}$$

where the average here is exactly the same as cases for LayerNorm applied to preactivations without residue connections. $l^\star$ labels some very large depth $l$.

*Remark* F.3.  For $\mu = 1$ case, the solution of (62) is

$$\mathcal{K}^l = \mathcal{K}^0 + \sum_{l'=1}^{l} \left( \frac{\sigma_w^2}{N_l} \sum_{j=1}^{N_l} \mathbb{E}_\theta \left[ \phi(\tilde{h}_j^{l'})\phi(\tilde{h}_j^{l'}) \right] + \sigma_b^2 \right) \,. \tag{65}$$

which is linearly growing since the expectation does not depend on depth. $\mathcal{K}^0$ is the NNGP kernel after the input layer.

**Theorem F.4.** *In the infinite width limit, the recurrence relation for Jacobians changes by a constant shift in the recursion coefficient.*

$$\mathcal{J}^{l_0,l+1} = \chi_{\mathcal{J}}^l \mathcal{J}^{l_0,l} \,, \tag{66}$$

*where for this case*

$$\chi_{\mathcal{J}}^l = \frac{\sigma_w^2}{N_l \mathcal{K}^l} \sum_{k=1}^{N_l} \mathbb{E}_\theta \left[ \phi'(\tilde{h}_k^l)\phi'(\tilde{h}_k^l) \right] + \mu^2 \,. \tag{67}$$

*Proof.*

$$\mathcal{J}^{l_0,l+1} = \frac{1}{N_{l+1}}\mathbb{E}_\theta\left[\sum_{i=1}^{N_{l+1}}\sum_{j=1}^{N_{l_0}}\frac{\partial h_i^{l+1}}{\partial h_j^{l_0}}\frac{\partial h_i^{l+1}}{\partial h_j^{l_0}}\right]$$

$$= \frac{1}{N_{l+1}}\mathbb{E}_\theta\left[\sum_{i=1}^{N_{l+1}}\sum_{j=1}^{N_{l_0}}\left(\sum_{k=1}^{N_l}\frac{\partial h_i^{l+1}}{\partial \tilde{h}_k^l}\frac{\partial \tilde{h}_k^l}{\partial h_k^l}\frac{\partial h_k^l}{\partial h_j^{l_0}}\right)\left(\sum_{m=1}^{N_l}\frac{\partial h_i^{l+1}}{\partial \tilde{h}_m^l}\frac{\partial \tilde{h}_m^l}{\partial h_m^l}\frac{\partial h_m^l}{\partial h_j^{l_0}}\right)\right]$$

$$= \frac{1}{N_{l+1}}\mathbb{E}_\theta\left[\sum_{i=1}^{N_{l+1}}\sum_{j=1}^{N_{l_0}}\sum_{k,m=1}^{N_l}\left(\frac{w_{ik}^{l+1}\phi'(\tilde{h}_k^l)}{\sqrt{\mathcal{K}^l}}+\mu\delta_{ik}\right)\left(\frac{w_{im}^{l+1}\phi'(\tilde{h}_m^l)}{\sqrt{\mathcal{K}^l}}+\mu\delta_{ik}\right)\left(\frac{\partial h_k^l}{\partial h_j^{l_0}}\frac{\partial h_m^l}{\partial h_j^{l_0}}\right)\right]$$

$$= \mathbb{E}_\theta\left[\left(\frac{\sigma_w^2}{N_l\mathcal{K}^l}\sum_{k=1}^{N_l}\phi'(\tilde{h}_k^l)\phi'(\tilde{h}_k^l)+\mu^2\right)\left(\sum_{j=1}^{N_{l_0}}\frac{\partial h_k^l}{\partial h_j^{l_0}}\frac{\partial h_k^l}{\partial h_j^{l_0}}\right)\right]$$

$$= \left(\frac{\sigma_w^2}{N_l\mathcal{K}^l}\sum_{k=1}^{N_l}\mathbb{E}_\theta\left[\phi'(\tilde{h}_k^l)\phi'(\tilde{h}_k^l)\right]+\mu^2\right)\mathcal{J}^{l_0,l}$$

$$= \chi_\mathcal{J}^l\mathcal{J}^{l_0,l}, \tag{68}$$

$\square$

*Remark* F.5. Assuming $l \geq l^\star$, we combine (64) and (67) to derive Equation (5) in the Theorem 1.3.

$$\chi_\mathcal{J}^* = (1-\mu^2)\frac{\frac{\sigma_w^2}{N_l}\sum_{k=1}^{N_l}\mathbb{E}_\theta\left[\phi'(\tilde{h}_k^l)\phi'(\tilde{h}_k^l)\right]}{\frac{\sigma_w^2}{N_l}\sum_{j=1}^{N_l}\mathbb{E}_\theta\left[\phi(\tilde{h}_j^l)\phi(\tilde{h}_j^l)\right]+\sigma_b^2}+\mu^2$$

$$= (1-\mu^2)\frac{A}{B}+\mu^2, \tag{69}$$

where $A = \sum_{k=1}^{N_l}\mathbb{E}_\theta\left[\phi'(\tilde{h}_k^l)\phi'(\tilde{h}_k^l)\right]$ and $B = \frac{\sigma_w^2}{N_l}\sum_{j=1}^{N_l}\mathbb{E}_\theta\left[\phi(\tilde{h}_j^l)\phi(\tilde{h}_j^l)\right]+\sigma_b^2$. Recall that $\xi = |\log\chi_\mathcal{J}^\star|^{-1}$. This directly leads us to Equation (5).

$$\xi = \frac{1}{|\log(\chi_\mathcal{J}^\star)|} = \frac{1}{|\log\left((1-\mu^2)\frac{A}{B}+\mu^2\right)|} \tag{70}$$

Note that in Equation (5), we discard the average over neurons in $A$ and $B$, since we are in the infinite width limit.

*Remark* F.6. As we mentioned above $\mu = 1$ needs extra care. Plugging $\mu = 1$ into the result (65) and (67) we find out that

$$\chi_\mathcal{J}^l\big|_{\mu=1} = \frac{\sigma_w^2\sum_{k=1}^{N_l}\mathbb{E}_\theta\left[\phi'(\tilde{h}_k^l)\phi'(\tilde{h}_k^l)\right]}{N_l\mathcal{K}^0+\sum_{l'=1}^l\left(\sigma_w^2\sum_{j=1}^{N_l}\mathbb{E}_\theta\left[\phi(\tilde{h}_j^l)\phi(\tilde{h}_j^l)\right]+N_l\sigma_b^2\right)}+1$$

$$\sim 1 + O\left(\frac{1}{l}\right). \tag{71}$$

Thus, we get everywhere criticality: critical behavior independent of the choice of initialization $(\sigma_b,\sigma_w)$. It also follows that the correlation length $\xi$ diverges in this case. (71) leads to power law behaviour in Jacobians (with exponent $\zeta$) at large depth. Note that the exponent $\zeta$ is not universal.

Remarks F.5 and F.6 together serve as the complete proof of Theorem 1.3

## G Critical Exponents

To prove theorem 2.7, we first need to find the critical exponent of the NNGP kernel [45].

**Lemma G.1.** *In the infinite width limit, consider a critically initialized network with an activation function $\phi$. The scaling behavior of the fluctuation $\delta\mathcal{K}^l \equiv \mathcal{K}^l - \mathcal{K}^\star$ in non-exponential. If the recurrence relation can be expand to leading order $\delta K^l$ as $\delta\mathcal{K}^{l+1} \approx \delta\mathcal{K}^l - c_n(\delta\mathcal{K}^l)^n$ for $n \geq 2$. The solution of $\delta\mathcal{K}^l$ is*

$$\delta\mathcal{K}^l = \frac{1}{c_n(n-1)} l^{-\zeta_\mathcal{K}}, \tag{72}$$

*where $\zeta_\mathcal{K} = \frac{1}{n-1}$.*

*Remark G.2.* The constant $c_n$ and the order of the first non-zero term $n$ is determined by the choice of activation function.

*Proof.* We can expand the recurrence relation for the NNGP kernel (9) to the second order of $\delta\mathcal{K}^l = \mathcal{K}^l - \mathcal{K}^\star$ on both sides.

$$\delta\mathcal{K}^{l+1} \approx \delta\mathcal{K}^l - c_n(\delta\mathcal{K}^l)^n. \tag{73}$$

Use power law ansatz $\delta\mathcal{K}^l = A\, l^{-\zeta_\mathcal{K}}$ then

$$(l+1)^{-\zeta_\mathcal{K}} = l^{-\zeta_\mathcal{K}} - c_n A\, l^{-n\zeta_\mathcal{K}}. \tag{74}$$

Multiply $l^{\zeta_\mathcal{K}}$ on both side then use Taylor expansion $(\frac{l}{l+1})^{\zeta_\mathcal{K}} \approx 1 - \frac{\zeta_\mathcal{K}}{l}$

$$\frac{\zeta_\mathcal{K}}{l} = c_n A l^{-(n-1)\zeta_\mathcal{K}}. \tag{75}$$

For arbitrary $l$, the only non-trivial solution of the equation above is

$$A = \frac{1}{c_n(n-1)} \quad \text{and} \quad \zeta_\mathcal{K} = \frac{1}{n-1}. \tag{76}$$

$\square$

*Proof of theorem 2.7.* We will assume $c_2 \neq 0$. Then use lemma G.1, we can expand $\chi^l_\mathcal{J}$ in terms of $\delta\mathcal{K}^l$. To leading order $l^{-1}$

$$\chi^l_\mathcal{J} \approx 1 - d_1\delta\mathcal{K}^l$$
$$= 1 - \frac{d_1}{c_2}l^{-1}. \tag{77}$$

Consider a sufficiently large $l$. In this case $O(l^{-1})$ approximation is valid. We write recurrence relations of Jacobians as

$$\mathcal{J}^{l_0,l} = \prod_{l'=l_0}^{l-1} \left(1 - \frac{d_1}{c_2}l'^{-1}\right) \mathcal{J}^{l_0,l_0}$$
$$\approx c_{l_0} \cdot l^{-\zeta}. \tag{78}$$

When $c_n = 0$ for all $n \geq 2$, from lemma G.1 we have $\delta\mathcal{K}^l = 0$. Thus the Jacobian saturates to some constant. $\square$

We checked the scaling empirically by plotting $\mathcal{J}^{0,l}$ vs. $l$ in a $\log$–$\log$ plot and fitting the slope. These results are presented in Fig.1.

# H  MLP-Mixer

In this section we would like to analyze an architecture called MLP-Mixer [48], which is based on multi-layer perceptrons (MLPs). A MLP-Mixer (i) chops images into patches, then applies affine transformations per patch, (ii) applies several Mixer Layers, (iii) applies pre-head LayerNorm, Global Average Pooling, and an output affine transformation. We will explain the architecture by showing forward pass equations.

Suppose one has a single input with dimension $(C_{in}, H_{in}, W_{in})$. We label it as $x_{\mu i}$, where the Greek letter labels channels and the Latin letter labels flattened pixels.

First of all the (i) is realized by a special convolutional layer, where kernel size $f$ is equal to the stride $s$. Then, the first convolution layer can be written as

$$h^0_{\mu i} = \sum_{j=1}^{f^2} \sum_{\nu=1}^{C_{in}} W^0_{\mu\nu;j} x_{\nu, j+(i-1)s^2} + b^0_{\mu i} , \tag{79}$$

where $f$ is the size of filter and $s$ is the stride. In our example $f = s$. Notice in PyTorch both bias and weights are sampled from a uniform distribution $\mathcal{U}(-\sqrt{k}, \sqrt{k})$, where $k = (C_{in} f^2)^{-1}$.

$$\mathbb{E}_\theta[W^0_{\mu\nu;i} W^0_{\rho\sigma;j}] = \frac{1}{3C_{in} f^2} \delta_{\mu\rho} \delta_{\nu\sigma} \delta_{ij} , \tag{80}$$

$$\mathbb{E}_\theta[b^0_{\mu i} b^0_{\nu j}] = \frac{1}{3C_{in} f^2} \delta_{\mu\nu} \delta_{ij} . \tag{81}$$

Notice that the output of Conv2d: $h^0_{\mu i} \in \mathbb{R}^{C \times N_p}$, where $C$ stands for channels and $N_p = H_{in} W_{in}/f^2$ stands for patches, both of them will be mixed later by Mixer layers.

Next, we stack $l$ Mixer Layers. A Mixer Layer contains LayerNorms and two MLPs, where the first one mixed patches $i, j$ (token mixing) with a hidden dimension $N_{tm}$, the second one mixed channels $\mu, \nu$ (channel-mixing) with a hidden dimension $N_{cm}$. Notice that for Mixer Layers we use the standard parameterization.

- First LayerNorm. It acts on channels $\mu$.

$$\tilde{h}^{6l}_{\mu i} = \frac{h^{6l}_{\mu i} - \mathbb{E}_C[h^{6l}_{\rho i}]}{\sqrt{\text{Var}_C[h^{6l}_{\rho i}]}} , \tag{82}$$

where we defined a channel mean $\mathbb{E}_C[h^{6l}_{\rho i}] \equiv \frac{1}{C} \sum_{\rho=1}^C h^{6l}_{\rho i}$ and channel variance $\text{Var}_C \equiv \mathbb{E}_C\left[\left(h^{6l}_{\rho i}\right)^2\right] - \left(\mathbb{E}_C[h^{6l}_{\rho i}]\right)^2$.

- First MlpBlock. It mixes patches $i, j$, preactivations from different channels that share the same weight and bias.
  - $6l + 1$: Linear Affine Layer.

$$h^{6l+1}_{\mu j} = \sum_{k=1}^{N_p} w^{6l+1}_{jk} \tilde{h}^{6l}_{\mu k} + b^{6l+1}_j . \tag{83}$$

  - $6l + 2$: Affine Layer.

$$h^{6l+2}_{\mu i} = \sum_{j=1}^{N_{tm}} w^{6l+2}_{ij} \phi(h^{6l+1}_{\mu j}) + b^{6l+2}_i , \tag{84}$$

where $N_{tm}$ stands for the hidden dimension of "token mixing".
  - $6l + 3$: Residual Connections.

$$h^{6l+3}_{\mu i} = h^{6l+2}_{\mu i} + \mu h^{6l}_{\mu i} . \tag{85}$$

- Second LayerNorm. It again acts on channels $\mu$.

$$\tilde{h}^{6l+3}_{\mu i} = \frac{h^{6l+3}_{\mu i} - \mathbb{E}_C[h^{6l+3}_{\rho i}]}{\sqrt{\text{Var}_C[h^{6l+3}_{\rho i}]}} . \tag{86}$$

- Second MlpBlock. It mixes channels $\mu, \nu$, preactivations from different patches share the same weight and bias.

– $6l + 4$: Linear Affine Layer.

$$h_{\nu i}^{6l+4} = \sum_{\rho=1}^{C} w_{\nu\rho}^{6l+4} \tilde{h}_{\rho i}^{6l+3} + b_{\nu}^{6l+4} \,. \tag{87}$$

– $6l + 5$. Affine Layer.

$$h_{\mu i}^{6l+5} = \sum_{\nu=1}^{N_{cm}} w_{\mu\nu}^{6l+5} \phi(h_{\nu i}^{6l+4}) + b_{\mu}^{6l+5} \,. \tag{88}$$

– $6l + 6$. Residual Connections.

$$h_{\mu i}^{6l+6} = h_{\mu i}^{6l+5} + \mu h_{\mu i}^{6l+3}. \tag{89}$$

Suppose the network has $L$ Mixer layers. After those layers the network has a pre-head LayerNorm layer, a global average pooling layer, and an output layer. The pre-head LayerNorm normalizes over channels $\mu$ can be described as the following

$$\tilde{h}_{\mu i}^{6L} = \frac{h_{\mu i}^{6L} - \mathbb{E}_C[h_{\rho i}^{6L}]}{\sqrt{\text{Var}_C[h_{\rho i}^{6L}]}} \,. \tag{90}$$

Global Average Pool over patches $i$.

$$h_{\mu}^{p} = \frac{1}{N_p} \sum_{i=1}^{N_p} \tilde{h}_{\mu i}^{6L} \,. \tag{91}$$

Output Layer

$$f_{\mu} = \sum_{\nu=1}^{C} w_{\mu\nu} h_{\nu}^{p} + b_{\mu} \,. \tag{92}$$

We plotted the phase diagram using the following quantity from repeating Mixer Layers:

$$\chi_{\mathcal{J}}^{\star} = \lim_{L \to \infty} \left( \frac{1}{N_p C} \sum_{i=1}^{N_p} \sum_{\mu=1}^{C} \mathbb{E}_{\theta} \left[ \sum_{\rho=1}^{C} \sum_{k=1}^{N_p} \frac{\partial h_{\mu i}^{6L}}{\partial h_{\rho k}^{6L-6}} \frac{\partial h_{\mu i}^{6L}}{\partial h_{\rho k}^{6L-6}} \right] \right) \,. \tag{93}$$

# I   Results for Scale Invariant Activation Functions

**Definition I.1** (Scale invariant activation functions)**.**

$$\phi(x) = a_+ \, x \, \Theta(x) + a_- \, x \, \Theta(-x) \,, \tag{94}$$

where $\Theta(x)$ is the Heaviside step function. ReLU is the special case with $a_+ = 1$ and $a_- = 0$.

## I.1   NNGP Kernel

First evaluate the average using lemma 2.2

$$\begin{aligned} \mathbb{E}_{\theta} \left[ \phi(h_i^l)\phi(h_i^l) \right] &= \frac{1}{\sqrt{2\pi \mathcal{K}^l}} \int dh_i^l \left( a_+^2 + a_-^2 \right) \left( h_i^l \right)^2 e^{-\frac{(h_i^l)^2}{2\mathcal{K}^l}} \\ &= \frac{a_+^2 + a_-^2}{2} \mathcal{K}^l \,. \end{aligned} \tag{95}$$

Thus we obtain the recurrence relation for the NNGP kernel with scale invariant activation function.

$$\mathcal{K}^{l+1} = \frac{\sigma_w^2 (a_+^2 + a_-^2)}{2} \mathcal{K}^l + \sigma_b^2 \,. \tag{96}$$

Finite fixed point of the recurrence relation above exists only if

$$\chi_{\mathcal{K}}^{\star} = \frac{\sigma_w^2(a_+^2 + a_-^2)}{2} \leq 1 \,.$$

(97)

As a result

$$\sigma_w^2 \leq \frac{2}{a_+^2 + a_-^2} \,.$$

(98)

For $\sigma_w^2 = \frac{2}{a_+^2 + a_-^2}$ case, finite fixed point exists only if $\sigma_b^2 = 0$.

## I.2   Jacobian(s)

The calculation is quite straightforward, by definition

$$\begin{aligned}
\chi_{\mathcal{J}}^l &= \sigma_w^2 \mathbb{E}_\theta \left[ \phi'(h_i^l) \phi'(h_i^l) \right] \\
&= \frac{\sigma_w^2}{\sqrt{2\pi\mathcal{K}^l}} \int dh_i^l \left[ a_+ \Theta(h_i^l) - a_- \Theta(h_i^l) \right]^2 e^{-\frac{(h_i^l)^2}{2\mathcal{K}^l}} \\
&= \frac{\sigma_w^2(a_+^2 + a_-^2)}{2} \,,
\end{aligned}$$

(99)

where we used the property $x\delta(x) = 0$ for Dirac's delta function to get the first line.

Thus the critical line is defined by

$$\sigma_w = \sqrt{\frac{2}{a_+^2 + a_-^2}} \,.$$

(100)

For ReLU with $a_+ = 1$ and $a_- = 0$, the network is at critical line when

$$\sigma_w = \sqrt{2} \,,$$

(101)

where the critical point is located at

$$(\sigma_w, \sigma_b) = (\sqrt{2}, 0) \,.$$

(102)

## I.3   Critical Exponents

Since the recurrence relations for the NNGP kernel and Jacobians are linear. Then from lemma G.1 and theorem 2.7

$$\zeta_{\mathcal{K}} = 0 \text{ and } \zeta = 0 \,.$$

(103)

## I.4   LayerNorm on Pre-activations

Use lemma D.9 and combine all known results for scale-invariant functions

$$\begin{aligned}
\chi_{\mathcal{J}}^l &= \frac{\sigma_w^2}{N_l \mathcal{K}^l} \sum_{k=1}^{N_l} \mathbb{E}_\theta \left[ \phi'(\tilde{h}_k^l) \phi'(\tilde{h}_k^l) \right] \Bigg|_{\tilde{\mathcal{K}}^{l-1} = 1} \\
&= \frac{\sigma_w^2(a_+^2 + a_-^2)}{\sigma_w^2(a_+^2 + a_-^2) + 2\sigma_b^2} \,.
\end{aligned}$$

(104)

In this case,

$$\chi_{\mathcal{J}}^l \leq 1$$

(105)

is always true. The equality only holds at $\sigma_b = 0$ line.

## I.5 LayerNorm on Activations

First we substitute $\mathcal{K}^{l-1} = \sigma_w^2 + \sigma_b^2$ into known results

$$\mathbb{E}_\theta \left[ \phi'(h_i^l)\phi'(h_i^l) \right] = \frac{a_+^2 + a_-^2}{2} \, , \tag{106}$$

$$\mathbb{E}_\theta \left[ \phi(h_i^l)\phi(h_i^l) \right] = \frac{a_+^2 + a_-^2}{2} (\sigma_w^2 + \sigma_b^2) \, . \tag{107}$$

There is a new expectation value we need to show explicitly

$$\begin{aligned}
\mathbb{E}_\theta \left[ \phi(h_i^l) \right] &= \frac{1}{\sqrt{2\pi(\sigma_w^2 + \sigma_b^2)}} \int_{-\infty}^{\infty} dh_i^l \phi(h_i^l) e^{-\frac{1}{2} h_i^l (\sigma_w^2 + \sigma_b^2)^{-1} h_i^l} \\
&= \frac{1}{\sqrt{2\pi(\sigma_w^2 + \sigma_b^2)}} \int_0^{\infty} dh_i^l (a_+ - a_-) h_i^l e^{-\frac{(h_i^l)^2}{2(\sigma_w^2 + \sigma_b^2)}} \\
&= (a_+ - a_-)\sqrt{\frac{\sigma_w^2 + \sigma_b^2}{2\pi}} \, . \tag{108}
\end{aligned}$$

Thus

$$\chi_{\mathcal{J}}^l = \frac{\sigma_w^2}{\sigma_w^2 + \sigma_b^2} \cdot \frac{\pi(a_+^2 + a_-^2)}{\pi(a_+^2 + a_-^2) - (a_+ - a_-)^2} \, . \tag{109}$$

The critical line is defined by $\chi_{\mathcal{J}}^\star = 1$, which can be solved as

$$\sigma_b = \sqrt{\frac{(a_+ - a_-)^2}{\pi(a_+^2 + a_-^2) - (a_+ - a_-)^2}} \sigma_w \, . \tag{110}$$

For ReLU with $a_+ = 1$ and $a_- = 0$

$$\begin{aligned}
\sigma_b &= \sqrt{\frac{1}{\pi - 1}} \sigma_w \\
&\approx 0.683 \sigma_w \, . \tag{111}
\end{aligned}$$

## I.6 Residual Connections

The recurrence relation for the NNGP kernel can be evaluated to be

$$\mathcal{K}^{l+1} = \frac{\sigma_w^2 (a_+^2 + a_-^2)}{2} \mathcal{K}^l + \sigma_b^2 + \mu^2 \mathcal{K}^l \, . \tag{112}$$

The condition for the existence of fixed point

$$\chi_{\mathcal{K}}^\star = \frac{\sigma_w^2 (a_+^2 + a_-^2)}{2} + \mu^2 \leq 1 \tag{113}$$

leads us to

$$\sigma_w^2 \leq \frac{2(1 - \mu^2)}{a_+^2 + a_-^2} \, . \tag{114}$$

For $\sigma_w^2 = \frac{2(1 - \mu^2)}{a_+^2 + a_-^2}$, finite fixed point exists only if $\sigma_b^2 = 0$. (Diverges linearly otherwise)

The recurrence coefficient for Jacobian is evaluated to be

$$\chi_{\mathcal{J}}^\star = \frac{\sigma_w^2 (a_+^2 + a_-^2)}{2} + \mu^2 \, . \tag{115}$$

The critical line is defined as

$$\sigma_w = \sqrt{\frac{2(1 - \mu^2)}{a_+^2 + a_-^2}} \, . \tag{116}$$

The critical point is located at $\left( \sqrt{\frac{2(1 - \mu^2)}{a_+^2 + a_-^2}}, 0 \right)$.

For ReLU, the critical point is at $\left( \sqrt{2(1 - \mu^2)}, 0 \right)$.

## I.7 Residual Connections with LayerNorm on Preactivations (Pre-LN)

Again use lemma D.9 and combine all known results for scale-invariant functions

$$
\begin{aligned}
\chi_{\mathcal{J}}^{\star} &= \lim_{l \to \infty} \left( \left. \frac{\sigma_w^2}{N_l \mathcal{K}^l} \sum_{k=1}^{N_l} \mathbb{E}_\theta \left[ \phi'(\tilde{h}_k^l) \phi'(\tilde{h}_k^l) \right] \right|_{\tilde{\mathcal{K}}^{l-1}=1} + \mu^2 \right) \\
&= \frac{\sigma_w^2 (a_+^2 + a_-^2)(1 - \mu^2)}{\sigma_w^2 (a_+^2 + a_-^2) + 2\sigma_b^2} + \mu^2 \\
&= 1 - \frac{2\sigma_b^2 (1 - \mu^2)}{\sigma_w^2 (a_+^2 + a_-^2) + 2\sigma_b^2}
\end{aligned}
\tag{117}
$$

Similar to the case without residue connections

$$
\chi_{\mathcal{J}}^l \leq 1
\tag{118}
$$

is always true. The equality only holds at $\sigma_b = 0$ line for $\mu < 1$.

Notice there is a very special case $\mu = 1$, where the whole $\sigma_b - \sigma_w$ plane is critical.

# J Results for erf Activation Function

**Definition J.1** (erf activation function).

$$
\phi(x) = \frac{2}{\sqrt{\pi}} \int_0^x e^{-t^2} dt \,.
\tag{119}
$$

## J.1 NNGP Kernel

To evaluate lemma 2.2 exactly, we introduce two dummy variables $\lambda_1$ and $\lambda_2$[51].

$$
\begin{aligned}
\mathbb{E}_\theta \left[ \phi(\lambda_1 h_i^l) \phi(\lambda_2 h_i^l) \right] &= \int d\lambda_1 \int d\lambda_2 \frac{d^2}{d\lambda_1 d\lambda_2} \mathbb{E}_\theta \left[ \phi(\lambda_1 h_i^l) \phi(\lambda_2 h_i^l) \right] \\
&= \int d\lambda_1 \int d\lambda_2 \int dh_i^l \frac{4}{\sqrt{2\pi^3 \mathcal{K}^l}} \left( h_i^l \right)^2 e^{-\left( \lambda_1^2 + \lambda_2^2 + \frac{1}{2\mathcal{K}^l} \right) \left( h_i^l \right)^2} \\
&= \int d\lambda_1 \int d\lambda_2 \frac{4\mathcal{K}^l}{\pi \left( 1 + 2\mathcal{K}^l (\lambda_1^2 + \lambda_2^2) \right)} \\
&= \frac{2}{\pi} \arcsin \left( \frac{2\mathcal{K}^l \lambda_1 \lambda_2}{1 + 2\mathcal{K}^l (\lambda_1^2 + \lambda_2^2)} \right) \,.
\end{aligned}
\tag{120}
$$

We use the special case where $\lambda_1 = \lambda_2 = 1$.

Thus the recurrence relation for the NNGP kernel with erf activation function is

$$
\mathcal{K}^{l+1} = \frac{2\sigma_w^2}{\pi} \arcsin \left( \frac{2\mathcal{K}^l}{1 + 2\mathcal{K}^l} \right) + \sigma_b^2 \,.
\tag{121}
$$

As in the scale-invariant case, finite fixed point only exists when

$$
\chi_{\mathcal{K}}^{\star} = \frac{4\sigma_w^2}{\pi} \frac{1}{(1 + 2\mathcal{K}^\star)\sqrt{1 + 4\mathcal{K}^\star}} \leq 1 \,.
\tag{122}
$$

Numerical results show the condition is satisfied everywhere in $\sigma_b - \sigma_w$ plane, where $\chi_{\mathcal{K}}^{\star} = 1$ is only possible when $\mathcal{K}^\star = 0$.

## J.2 Jacobians

Follow the definition

$$
\begin{aligned}
\chi_{\mathcal{J}}^l &= \sigma_w^2 \mathbb{E}_\theta \left[ \phi'(h_i^l)\phi'(h_i^l) \right] \\
&= \frac{4\sigma_w^2}{\sqrt{2\pi^3 \mathcal{K}^l}} \int dh_i^l \, e^{-2(h_i^l)^2} \, e^{-\frac{(h_i^l)^2}{2\mathcal{K}^l}} \\
&= \frac{4\sigma_w^2}{\pi} \frac{1}{\sqrt{1+4\mathcal{K}^l}} \, .
\end{aligned}
\tag{123}
$$

To find phase boundary $\chi_{\mathcal{J}}^\star = 1$, we need to combine Eq.(121) and Eq.(123) and evaluate them at $\mathcal{K}^\star$.

$$
\mathcal{K}^\star = \frac{2\sigma_w^2}{\pi} \arcsin\left(\frac{2\mathcal{K}^\star}{1+2\mathcal{K}^\star}\right) + \sigma_b^2 \, ,
\tag{124}
$$

$$
\chi_{\mathcal{J}}^\star = \frac{4\sigma_w^2}{\pi} \frac{1}{\sqrt{1+4\mathcal{K}^\star}} = 1 \, .
\tag{125}
$$

One can solve the equations above and find the critical line

$$
\sigma_b = \sqrt{\frac{16\sigma_w^4 - \pi^2}{4\pi^2} - \frac{2\sigma_w^2}{\pi} \arcsin\left(\frac{16\sigma_w^4 - \pi^2}{16\sigma_w^4 + \pi^2}\right)} \, .
\tag{126}
$$

Critical point is reached by further requiring $\chi_{\mathcal{K}}^\star = 1$. Since $\chi_{\mathcal{K}}^\star \le \chi_{\mathcal{J}}^\star$, the only possible case is $\mathcal{K}^\star = 0$, which is located at

$$
(\sigma_w, \sigma_b) = \left(\sqrt{\frac{\pi}{4}}, 0\right) \, .
\tag{127}
$$

## J.3 Critical Exponents

We show how to extract critical exponents of the NNGP kernel and Jacobians of erf activation function.

Critical point for erf is at $(\sigma_b, \sigma_w) = (0, \sqrt{\frac{\pi}{4}})$, with $\mathcal{K}^\star = 0$. Now suppose $l$ is large enough such that the deviation of $\mathcal{K}^l$ from fixed point value $\mathcal{K}^\star$ is small. Define $\delta\mathcal{K}^l \equiv \mathcal{K}^l - \mathcal{K}^\star$. Eq.(121) can be rewritten as

$$
\begin{aligned}
\delta\mathcal{K}^{l+1} &= \frac{1}{2} \arcsin\left(\frac{2\delta\mathcal{K}^l}{1+2\delta\mathcal{K}^l}\right) \\
&\approx \delta\mathcal{K}^l - 2(\delta\mathcal{K}^l)^2 \, .
\end{aligned}
\tag{128}
$$

From lemma G.1

$$
A = \frac{1}{2} \text{ and } \zeta_{\mathcal{K}} = 1 \, .
\tag{129}
$$

Next we analyze critical exponent of Jacobians by expanding (123) around $\mathcal{K}^\star = 0$ critical point $(\sigma_b, \sigma_w) = (0, \sqrt{\frac{\pi}{4}})$.

To leading order $l^{-1}$ we have

$$
\begin{aligned}
\chi_{\mathcal{J}}^l &\approx 1 - 2\delta K^l \\
&\approx 1 - \frac{1}{l} \, .
\end{aligned}
\tag{130}
$$

Thus the recurrence relation for partial Jacobian, at large $l$, takes form

$$
\mathcal{J}^{l_0, l+1} = \left(1 - \frac{1}{l}\right) \mathcal{J}^{l_0, l} \, .
\tag{131}
$$

At large $l$

$$\mathcal{J}^{l_0, l} = c_{l_0} \, l^{-1} \,, \tag{132}$$

with a non-universal constant $c_{l_0}$.

The critical exponent is

$$\zeta = 1 \,, \tag{133}$$

which is the same as $\zeta_{\mathcal{K}}$.

## J.4 LayerNorm on Pre-activations

Use lemma D.9, we have

$$
\begin{aligned}
\chi_{\mathcal{J}}^l &= \frac{\sigma_w^2}{N_l \mathcal{K}^l} \sum_{k=1}^{N_l} \mathbb{E}_\theta \left[ \phi'(\tilde{h}_k^l) \phi'(\tilde{h}_k^l) \right] \Bigg|_{\tilde{\mathcal{K}}^{l-1} = 1} \\
&= \frac{4\sigma_w^2}{\sqrt{5} \left[ 2\sigma_w^2 \arcsin\left(\frac{2}{3}\right) + \pi \sigma_b^2 \right]} \,.
\end{aligned}
\tag{134}
$$

The critical line is then defined by

$$
\begin{aligned}
\sigma_b &= \sqrt{\frac{2}{\pi} \left[ \frac{2}{\sqrt{5}} - \arcsin\left(\frac{2}{3}\right) \right]} \, \sigma_w \\
&\approx 0.324 \sigma_w \,.
\end{aligned}
\tag{135}
$$

## J.5 LayerNorm on Activations

Due to the symmetry of erf activation function $\mathbb{E}_\theta \left[ \phi(h_i^l) \right] = 0$, we only need to modify our known results.

$$\mathbb{E}_\theta \left[ \phi'(h_i^l) \phi'(h_i^l) \right] = \frac{4}{\pi} \frac{1}{\sqrt{1 + 4(\sigma_w^2 + \sigma_b^2)}} \,, \tag{136}$$

$$\mathbb{E}_\theta \left[ \phi(h_i^l) \phi(h_i^l) \right] = \frac{2}{\pi} \arcsin\left( \frac{2(\sigma_w^2 + \sigma_b^2)}{1 + 2(\sigma_w^2 + \sigma_b^2)} \right) \,. \tag{137}$$

Thus

$$\chi_{\mathcal{J}}^l = \frac{2\sigma_w^2}{\sqrt{1 + 4(\sigma_w^2 + \sigma_b^2)}} \cdot \frac{1}{\arcsin\left( \frac{2(\sigma_w^2 + \sigma_b^2)}{1 + 2(\sigma_w^2 + \sigma_b^2)} \right)} \,, \tag{138}$$

where the phase boundary is defined by the transcendental equation $\chi_{\mathcal{J}}^l = 1$.

## J.6 Residual Connections

The recurrence relation for the NNGP kernel can be evaluated to be

$$\mathcal{K}^{l+1} = \frac{2\sigma_w^2}{\pi} \arcsin\left( \frac{2\mathcal{K}^l}{1 + 2\mathcal{K}^l} \right) + \sigma_b^2 + \mu^2 \mathcal{K}^l \,. \tag{139}$$

Finite fixed point only exists when

$$\chi_{\mathcal{K}}^\star = \frac{4\sigma_w^2}{\pi} \frac{1}{(1 + 2\mathcal{K}^\star)\sqrt{1 + 4\mathcal{K}^\star}} + \mu^2 \leq 1 \,. \tag{140}$$

Notice that $\chi_{\mathcal{K}}^\star \leq \chi_{\mathcal{J}}^\star$ still holds, where the equality holds only when $\mathcal{K}^\star = 0$.

The recurrence coefficient for Jacobian is evaluated to be

$$\chi_{\mathcal{J}}^\star = \frac{4\sigma_w^2}{\pi} \frac{1}{\sqrt{1 + 4\mathcal{K}^\star}} + \mu^2 \,. \tag{141}$$

The critical line is defined as

$$\sigma_b = \sqrt{\frac{16\sigma_w^4 - \pi^2(1-\mu^2)^2}{4\pi^2(1-\mu^2)} - \frac{2\sigma_w^2}{\pi}\arcsin\left(\frac{16\sigma_w^4 - \pi^2(1-\mu^2)^2}{16\sigma_w^4 + \pi^2(1-\mu^2)^2}\right)}. \tag{142}$$

Critical point is reached by further requiring $\chi_{\mathcal{K}}^\star = 1$. Since $\chi_{\mathcal{K}}^\star \leq \chi_{\mathcal{J}}^\star$, the only possible case is $\mathcal{K}^\star = 0$, which is located at

$$(\sigma_w, \sigma_b) = \left(\sqrt{\frac{\pi(1-\mu^2)}{4}}, 0\right). \tag{143}$$

Note that for $\mu = 1$, one needs to put extra effort into analyzing the scaling behavior. First we notice that $\mathcal{K}^l$ monotonically increases with depth $l$ – the recurrence relation for the NNGP kernel at large $l$ (or large $\mathcal{K}^l$) is

$$\mathcal{K}^{l+1} \approx \sigma_w^2 + \sigma_b^2 + \mathcal{K}^l, \tag{144}$$

which regulates the first term in (141).

For $\mu = 1$ at large depth

$$\chi_{\mathcal{J}}^l \sim 1 + \frac{4\sigma_w^2}{\pi\sqrt{C_0 + 4(\sigma_w^2 + \sigma_b^2)l}}. \tag{145}$$

Here $C_0$ is a constant that depends on the input.

We can approximate the asymptotic form of $\log \mathcal{J}^{l_0,l}$ as follows

$$\begin{aligned}
\log \mathcal{J}^{l_0,l} &= \log\left(\prod_{l'=l_0}^{l} \chi_{\mathcal{J}}^{l'}\right) \\
&= \sum_{l'=l_0}^{l} \log\left(1 + \frac{4\sigma_w^2}{\pi\sqrt{C_0 + 4(\sigma_w^2 + \sigma_b^2)l'}}\right) \\
&\approx \int_{l_0}^{l} dl' \log\left(1 + \frac{4\sigma_w^2}{\pi\sqrt{C_0 + 4(\sigma_w^2 + \sigma_b^2)l'}}\right) \\
&\sim 2\tilde{c}\sqrt{l} + O(\log l),
\end{aligned} \tag{146}$$

where $\tilde{c} = \frac{2\sigma_w^2}{\pi\sqrt{\sigma_w^2 + \sigma_b^2}}$.

We conclude that at large depth, the APJN for $\mu = 1$, erf networks can be written as

$$\mathcal{J}^{l_0,l} \sim O\left(e^{2\tilde{c}\sqrt{l} + O(\log l)}\right). \tag{147}$$

This result checks out empirically, as shown in Figure 9.[8]

## J.7 Residual Connections with LayerNorm on Preactivations (Pre-LN)

Use lemma D.9 and the results we had without residue connections for erf with LayerNorm on preactivations.

$$\begin{aligned}
\chi_{\mathcal{J}}^* &= \lim_{l\to\infty} \left(\frac{\sigma_w^2}{N_l\mathcal{K}^l}\sum_{k=1}^{N_l}\mathbb{E}_\theta\left[\phi'(\tilde{h}_k^l)\phi'(\tilde{h}_k^l)\right]\Bigg|_{\tilde{\mathcal{K}}^{l-1}=1} + \mu^2\right) \\
&= \frac{4\sigma_w^2(1-\mu^2)}{\sqrt{5}\left[2\sigma_w^2\arcsin\left(\frac{2}{3}\right) + \pi\sigma_b^2\right]} + \mu^2.
\end{aligned} \tag{148}$$

The critical line is then defined by

$$\sigma_b = \sqrt{\frac{2}{\pi}\left[\frac{2}{\sqrt{5}} - \arcsin\left(\frac{2}{3}\right)\right]}\sigma_w \tag{149}$$

$$\approx 0.324\sigma_w.$$

---

[8]We used NTK parameterization for this experiment. However, we emphasize that it does not affect the final result.

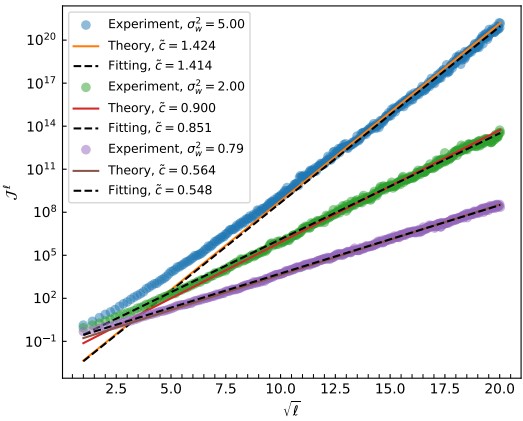

Figure 9: $\log(\mathcal{J}^{l_0,l})$-$\sqrt{l}$ for $\mu = 1$, $\sigma_b^2 = 0$, erf.

# K  Results for GELU Activation Function

**Definition K.1** (GELU activation function).

$$\phi(x) = \frac{x}{2}\left[1 + \text{erf}\left(\frac{x}{\sqrt{2}}\right)\right]$$

$$= \frac{x}{2}\left[1 + \frac{2}{\sqrt{\pi}}\int_0^{\frac{x}{\sqrt{2}}} e^{-t^2}\,dt\right]. \tag{150}$$

## K.1  NNGP Kernel

Use lemma 2.2 for GELU

$$
\begin{aligned}
\mathbb{E}_\theta\left[\phi(h_i^l)\phi(h_i^l)\right] &= \frac{1}{\sqrt{2\pi\mathcal{K}^l}}\int dh_i^l \frac{(h_i^l)^2}{4}\left[1 + \text{erf}\left(\frac{h_i^l}{\sqrt{2}}\right)\right]^2 e^{-\frac{(h_i^l)^2}{2\mathcal{K}^l}}\\
&= \frac{1}{\sqrt{2\pi\mathcal{K}^l}}\int dh_i^l \frac{(h_i^l)^2}{4}\left[1 + \text{erf}^2\left(\frac{h_i^l}{\sqrt{2}}\right)\right] e^{-\frac{(h_i^l)^2}{2\mathcal{K}^l}}\\
&= \frac{\mathcal{K}^l}{4} + \frac{1}{\sqrt{32\pi\mathcal{K}^l}}\int dh_i^l\,(h_i^l)^2\text{erf}^2\left(\frac{h_i^l}{\sqrt{2}}\right) e^{-\frac{(h_i^l)^2}{2\mathcal{K}^l}}\\
&= \frac{\mathcal{K}^l}{4} + \frac{\mathcal{K}^l}{\sqrt{32\pi\mathcal{K}^l}}\int dh_i^l\,\text{erf}^2\left(\frac{h_i^l}{\sqrt{2}}\right) e^{-\frac{(h_i^l)^2}{2\mathcal{K}^l}}\\
&\quad + \frac{(\mathcal{K}^l)^2}{\sqrt{32\pi\mathcal{K}^l}}\int dh_i^l\left[\text{erf}'\left(\frac{h_i^l}{\sqrt{2}}\right)\text{erf}'\left(\frac{h_i^l}{\sqrt{2}}\right) + \text{erf}\left(\frac{h_i^l}{\sqrt{2}}\right)\text{erf}''\left(\frac{h_i^l}{\sqrt{2}}\right)\right] e^{-\frac{(h_i^l)^2}{2\mathcal{K}^l}}\\
&= \frac{\mathcal{K}^l}{4} + \frac{\mathcal{K}^l}{2\pi}\left[\arcsin\left(\frac{\mathcal{K}^l}{1+\mathcal{K}^l}\right) + \frac{2\mathcal{K}^l}{(1+\mathcal{K}^l)\sqrt{1+2\mathcal{K}^l}}\right], \tag{151}
\end{aligned}
$$

where from the third line to the fourth line we used integrate by parts twice, and to get the last line we used results from erf activations.

Thus the recurrence relation for the NNGP kernel is

$$\mathcal{K}^{l+1} = \left[\frac{\mathcal{K}^l}{4} + \frac{\mathcal{K}^l}{2\pi}\arcsin\left(\frac{\mathcal{K}^l}{1+\mathcal{K}^l}\right) + \frac{(\mathcal{K}^l)^2}{\pi(1+\mathcal{K}^l)\sqrt{1+2\mathcal{K}^l}}\right]\sigma_w^2 + \sigma_b^2. \tag{152}$$

As a result

$$\chi_\mathcal{K}^\star = \frac{\sigma_w^2}{4} + \frac{\sigma_w^2}{2\pi}\left[\arcsin\left(\frac{\mathcal{K}^\star}{1+\mathcal{K}^\star}\right) + \frac{4(\mathcal{K}^\star)^3 + 11(\mathcal{K}^\star)^2 + 5\mathcal{K}^\star}{(1+\mathcal{K}^\star)^2(1+2\mathcal{K}^\star)^{\frac{3}{2}}}\right]. \tag{153}$$

## K.2 Jacobians

Follow the definition

$$\chi_{\mathcal{J}}^l = \sigma_w^2 \mathbb{E}_\theta \left[ \phi'(h_i^l) \phi'(h_i^l) \right]$$

$$= \frac{\sigma_w^2}{\sqrt{2\pi \mathcal{K}^l}} \int dh_i^l \left[ \frac{1}{2} + \frac{1}{2} \mathrm{erf} \left( \frac{h_i^l}{\sqrt{2}} \right) + \frac{e^{-\frac{(h_i^l)^2}{2}} h_i^l}{\sqrt{2\pi}} \right]^2 e^{-\frac{(h_i^l)^2}{2\mathcal{K}^l}}$$

$$= \frac{\sigma_w^2}{\sqrt{2\pi \mathcal{K}^l}} \int dh_i^l \left[ \frac{1}{4} + \frac{1}{4} \mathrm{erf} \left( \frac{h_i^l}{\sqrt{2}} \right)^2 + \frac{h_i^l \mathrm{erf} \left( \frac{h_i^l}{\sqrt{2}} \right) e^{-\frac{(h_i^l)^2}{2}}}{\sqrt{2\pi}} + \frac{e^{-(h_i^l)^2} (h_i^l)^2}{2\pi} \right] e^{-\frac{(h_i^l)^2}{2\mathcal{K}^l}}$$

$$= \frac{\sigma_w^2}{4} + \frac{\sigma_w^2}{2\pi} \left[ \arcsin \left( \frac{\mathcal{K}^l}{1 + \mathcal{K}^l} \right) + \frac{\mathcal{K}^l (3 + 5\mathcal{K}^l)}{(1 + \mathcal{K}^l)(1 + 2\mathcal{K}^l)^{\frac{3}{2}}} \right] , \tag{154}$$

where we dropped odd function terms to get the third line, and to get the last line we used known result for erf in the second term, integration by parts in the third term.

Here to get the critical line is harder. One can use the recurrence relation for the NNGP kernel at fixed point $\mathcal{K}^\star$ and $\chi_{\mathcal{J}}^\star = 1$

$$\mathcal{K}^\star = \frac{\sigma_w^2}{4} \mathcal{K}^\star + \frac{\sigma_w^2}{2\pi} \left[ \arcsin \left( \frac{\mathcal{K}^\star}{1 + \mathcal{K}^\star} \right) + \frac{\sigma_w^2 \mathcal{K}^\star}{\pi (1 + \mathcal{K}^\star)\sqrt{1 + 2\mathcal{K}^\star}} \right] \mathcal{K}^\star + \sigma_b^2 , \tag{155}$$

$$\chi_{\mathcal{J}}^\star = \frac{\sigma_w^2}{4} + \frac{\sigma_w^2}{2\pi} \left[ \arcsin \left( \frac{\mathcal{K}^\star}{1 + \mathcal{K}^\star} \right) + \frac{\mathcal{K}^\star (3 + 5\mathcal{K}^\star)}{(1 + \mathcal{K}^\star)(1 + 2\mathcal{K}^\star)^{\frac{3}{2}}} \right] = 1 . \tag{156}$$

Cancel the $\arcsin$ term, $\sigma_w$ and $\sigma_b$ then can be written as a function of $\mathcal{K}^\star$

$$\sigma_w = 2 \left[ 1 + \frac{2\mathcal{K}^\star (3 + 5\mathcal{K}^\star)}{\pi (1 + \mathcal{K}^\star)(1 + 2\mathcal{K}^\star)^{\frac{3}{2}}} + \frac{2}{\pi} \arcsin \left( \frac{\mathcal{K}^\star}{1 + \mathcal{K}^\star} \right) \right]^{-\frac{1}{2}} , \tag{157}$$

$$\sigma_b = \frac{\mathcal{K}^\star}{\sqrt{2\pi}(1 + 2\mathcal{K}^\star)^{\frac{3}{4}}} \sigma_w . \tag{158}$$

One can then scan $\mathcal{K}^\star$ to draw the critical line.

In order to locate the critical point, we further require $\chi_{\mathcal{K}}^\star = 1$. To locate the critical point, we solve $\chi_{\mathcal{J}}^\star - \chi_{\mathcal{K}}^\star = 0$ instead. We have

$$\frac{\sigma_w^2 [(\mathcal{K}^\star)^3 - 3(\mathcal{K}^\star)^2 - 2\mathcal{K}^\star]}{2\pi (1 + \mathcal{K}^\star)^2 (1 + 2\mathcal{K}^\star)^{\frac{3}{2}}} = 0 , \tag{159}$$

which has two non-negative solutions out of three

$$\mathcal{K}^\star = 0 \text{ and } \mathcal{K}^\star = \frac{3 + \sqrt{17}}{2} . \tag{160}$$

One can then solve $\sigma_b$ and $\sigma_w$ by plugging corresponding $K^\star$ values.

$$(\sigma_w, \sigma_b) = (2, 0) , \text{ for } \mathcal{K}^\star = 0 , \tag{161}$$

$$(\sigma_w, \sigma_b) \approx (1.408, 0.416) , \text{ for } \mathcal{K}^\star = \frac{3 + \sqrt{17}}{2} . \tag{162}$$

## K.3 Critical Exponents

GELU behaves in a different way compared to erf. First we discuss the $\mathcal{K}^\star = 0$ critical point, which is located at $(\sigma_b, \sigma_w) = (0, 2)$. We expand Eq.(152), and keep next to leading order $\delta \mathcal{K}^l = \mathcal{K}^l - \mathcal{K}^\star$

$$\delta \mathcal{K}^{l+1} \approx \delta \mathcal{K}^l + \frac{6}{\pi} (\delta \mathcal{K}^l)^2 . \tag{163}$$

From lemma G.1

$$A = -\frac{\pi}{6} \text{ and } \zeta_{\mathcal{K}} = 1 \,, \tag{164}$$

which is not possible since $\delta \mathcal{K}^l \geq 0$ for this case. This result means scaling analysis is not working here.

Next, we consider the other fixed point with $\mathcal{K}^\star = \frac{3+\sqrt{17}}{2}$ at $(\sigma_b, \sigma_w) = (0.416, 1.408)$. Expand the NNGP kernel recurrence relation again.

$$\delta \mathcal{K}^{l+1} \approx \delta \mathcal{K}^l + 0.00014 (\delta \mathcal{K}^l)^2 \,. \tag{165}$$

Following the same analysis, we find

$$\delta \mathcal{K}^l \approx -7142.9 \, l^{-1} \,. \tag{166}$$

Looks like scaling analysis works for this case, since $\mathcal{K}^\star > 0$. The solution shows that the critical point is half-stable[45]. If $\mathcal{K}^l < \mathcal{K}^\star$, the fixed point is repealing, while when $\mathcal{K}^l > \mathcal{K}^\star$, the fixed point is attractive. However, the extremely large coefficient in the scaling behavior of $\delta \mathcal{K}^l$ embarrasses the analysis. Since for any network with a reasonable depth, the deviation $\delta \mathcal{K}^l$ is not small.

Now we can expand $\chi_{\mathcal{J}}^l$ at some large depth, up to leading order $l^{-1}$.

$$\chi_{\mathcal{J}}^l \approx 1 - \frac{66.668}{l} \,. \tag{167}$$

Then

$$\delta \mathcal{J}^{l_0, l} \approx c_{l_0} l^{-66.668} \,, \tag{168}$$

where $c_{l_0}$ is a positive non-universal constant.

Critical exponent

$$\zeta = 66.668 \,. \tag{169}$$

Which in practice is not traceable.

### K.4 LayerNorm on Pre-activations

Use lemma D.9, we have

$$\begin{aligned}
\chi_{\mathcal{J}}^l &= \frac{\sigma_w^2}{N_l \mathcal{K}^l} \sum_{k=1}^{N_l} \mathbb{E}_\theta \left[ \phi'(\tilde{h}_k^l) \phi'(\tilde{h}_k^l) \right] \Bigg|_{\tilde{\mathcal{K}}^{l-1}=1} \\
&= \frac{\sigma_w^2 (6\pi + 4\sqrt{3})}{\sigma_w^2 (6\pi + 3\sqrt{3}) + 18\pi\sigma_b^2} \,.
\end{aligned} \tag{170}$$

The critical line is then at

$$\begin{aligned}
\sigma_b &= \left( 6\sqrt{3}\pi \right)^{-\frac{1}{2}} \sigma_w \\
&\approx 0.175 \sigma_w \,.
\end{aligned} \tag{171}$$

### K.5 LayerNorm on Activations

First, we need to evaluate a new expectation value

$$\begin{aligned}
\mathbb{E}_\theta \left[ \phi(h_i^l) \right] &= \frac{1}{\sqrt{2\pi(\sigma_w^2 + \sigma_b^2)}} \int dh_i^l \frac{h_i^l}{2} \left[ 1 + \text{erf}\left( \frac{x}{\sqrt{2}} \right) \right] e^{-\frac{(h_i^l)^2}{2(\sigma_w^2 + \sigma_b^2)}} \\
&= \frac{\sigma_w^2 + \sigma_b^2}{\sqrt{2\pi(1 + \sigma_w^2 + \sigma_b^2)}} \,,
\end{aligned} \tag{172}$$

where we used integration by parts to get the result.

The other integrals are modified to

$$\mathbb{E}_\theta\left[\phi'(h_i^l)\phi'(h_i^l)\right] = \frac{1}{4} + \frac{1}{2\pi}\left[\arcsin\left(\frac{\sigma_w^2 + \sigma_b^2}{1 + \sigma_w^2 + \sigma_b^2}\right) + \frac{(\sigma_w^2 + \sigma_b^2)[3 + 5(\sigma_w^2 + \sigma_b^2)]}{(1 + \sigma_w^2 + \sigma_b^2)[1 + 2(\sigma_w^2 + \sigma_b^2)]^{\frac{3}{2}}}\right],$$
(173)

$$\mathbb{E}_\theta\left[\phi(h_i^l)\phi(h_i^l)\right] = \frac{\sigma_w^2 + \sigma_b^2}{4} + \frac{\sigma_w^2 + \sigma_b^2}{2\pi}\arcsin\left(\frac{\sigma_w^2 + \sigma_b^2}{1 + \sigma_w^2 + \sigma_b^2}\right) + \frac{(\sigma_w^2 + \sigma_b^2)^2}{\pi(1 + \sigma_w^2 + \sigma_b^2)\sqrt{1 + 2(\sigma_w^2 + \sigma_b^2)}}.$$
(174)

One can then combine those results to find $\chi_{\mathcal{J}}^l$

$$\chi_{\mathcal{J}}^l = \frac{\sigma_w^2\left(1 + \sigma_w^2 + \sigma_b^2\right)\left[\pi + 2\arcsin\left(\frac{\sigma_w^2 + \sigma_b^2}{1 + \sigma_w^2 + \sigma_b^2}\right) + \frac{2(\sigma_w^2 + \sigma_b^2)(3 + 5(\sigma_w^2 + \sigma_b^2))}{(1 + \sigma_w^2 + \sigma_b^2)(1 + 2(\sigma_w^2 + \sigma_b^2))^{\frac{3}{2}}}\right]}{\pi(\sigma_w^2 + \sigma_b^2)(1 + \sigma_w^2 + \sigma_b^2) - 2(\sigma_w^2 + \sigma_b^2)^2 + \frac{4(\sigma_w^2 + \sigma_b^2)^2}{\sqrt{1 + 2(\sigma_w^2 + \sigma_b^2)}} + 2(\sigma_w^2 + \sigma_b^2)(1 + \sigma_w^2 + \sigma_b^2)\arcsin\left(\frac{\sigma_w^2 + \sigma_b^2}{1 + \sigma_w^2 + \sigma_b^2}\right)}.$$
(175)

The critical line defined by $\chi_{\mathcal{J}}^l = 1$, one can numerically solve it by scanning over $\sigma_b$ and $\sigma_w$.

## K.6 Residual Connections

The recurrence relation for the NNGP kernel is

$$\mathcal{K}^{l+1} = \left[\frac{\mathcal{K}^l}{4} + \frac{\mathcal{K}^l}{2\pi}\arcsin\left(\frac{\mathcal{K}^l}{1 + \mathcal{K}^l}\right) + \frac{(\mathcal{K}^l)^2}{\pi(1 + \mathcal{K}^l)\sqrt{1 + 2\mathcal{K}^l}}\right]\sigma_w^2 + \sigma_b^2 + \mu^2\mathcal{K}^l.$$
(176)

Fixed point exists if

$$\chi_{\mathcal{K}}^\star = \frac{\sigma_w^2}{4} + \frac{\sigma_w^2}{2\pi}\left[\arcsin\left(\frac{\mathcal{K}^\star}{1 + \mathcal{K}^\star}\right) + \frac{4(\mathcal{K}^\star)^3 + 11(\mathcal{K}^\star)^2 + 5\mathcal{K}^\star}{(1 + \mathcal{K}^\star)^2(1 + 2\mathcal{K}^\star)^{\frac{3}{2}}}\right] + \mu^2 \leq 1.$$
(177)

The recurrence coefficient for Jacobian is

$$\chi_{\mathcal{J}}^\star = \frac{\sigma_w^2}{4} + \frac{\sigma_w^2}{2\pi}\left[\arcsin\left(\frac{\mathcal{K}^\star}{1 + \mathcal{K}^\star}\right) + \frac{\mathcal{K}^\star(3 + 5\mathcal{K}^\star)}{(1 + \mathcal{K}^\star)(1 + 2\mathcal{K}^\star)^{\frac{3}{2}}}\right] + \mu^2.$$
(178)

Phase boundary is shifted

$$\sigma_w = 2\sqrt{1 - \mu^2}\left[1 + \frac{2\mathcal{K}^\star(3 + 5\mathcal{K}^\star)}{\pi(1 + \mathcal{K}^\star)(1 + 2\mathcal{K}^\star)^{\frac{3}{2}}} + \frac{2}{\pi}\arcsin\left(\frac{\mathcal{K}^\star}{1 + \mathcal{K}^\star}\right)\right]^{-\frac{1}{2}},$$
(179)

$$\sigma_b = \frac{\mathcal{K}^\star}{\sqrt{2\pi}(1 + 2\mathcal{K}^\star)^{\frac{3}{4}}}\sigma_w.$$
(180)

One can again scan over $\mathcal{K}^\star$ to draw the critical line.

In order to locate the critical point, we further require $\chi_{\mathcal{K}}^\star = 1$. To locate the critical point, we solve $\chi_{\mathcal{J}}^\star - \chi_{\mathcal{K}}^\star = 0$ instead. We have

$$\frac{\sigma_w^2[(\mathcal{K}^\star)^3 - 3(\mathcal{K}^\star)^2 - 2\mathcal{K}^\star]}{2\pi(1 + \mathcal{K}^\star)^2(1 + 2\mathcal{K}^\star)^{\frac{3}{2}}} = 0,$$
(181)

which has two non-negative solutions out of three

$$\mathcal{K}^\star = 0 \text{ and } \mathcal{K}^\star = \frac{3 + \sqrt{17}}{2}.$$
(182)

One can then solve $\sigma_b$ and $\sigma_w$ by plugging corresponding $K^\star$ values.

$$(\sigma_w, \sigma_b) = (2\sqrt{1 - \mu^2}, 0), \text{ for } \mathcal{K}^\star = 0,$$
(183)

$$(\sigma_w, \sigma_b) \approx (1.408\sqrt{1 - \mu^2}, 0.416\sqrt{1 - \mu^2}), \text{ for } \mathcal{K}^\star = \frac{3 + \sqrt{17}}{2}.$$
(184)

## K.7 Residual Connections with LayerNorm on Preactivations (Pre-LN)

Use lemma D.9 and the results we had without residue connections for GELU.

$$\chi^*_{\mathcal{J}} = \lim_{l\to\infty} \left( \frac{\sigma_w^2}{N_l\mathcal{K}^l} \sum_{k=1}^{N_l} \mathbb{E}_\theta\left[\phi'(\tilde{h}_k^l)\phi'(\tilde{h}_k^l)\right]\Bigg|_{\tilde{\mathcal{K}}^{l-1}=1} + \mu^2 \right)$$

$$= \frac{\sigma_w^2(6\pi + 4\sqrt{3})(1-\mu^2)}{\sigma_w^2(6\pi + 3\sqrt{3}) + 18\pi\sigma_b^2} + \mu^2$$

$$= 1 - \frac{(\sqrt{3}\sigma_w^2 - 18\pi\sigma_b^2)(1-\mu^2)}{\sigma_w^2(6\pi + 3\sqrt{3}) + 18\pi\sigma_b^2}. \tag{185}$$

The critical line is then at

$$\sigma_b = \left(6\sqrt{3}\pi\right)^{-\frac{1}{2}} \sigma_w \tag{186}$$

$$\approx 0.175\sigma_w,$$

just like without residue connections.

# L   Additional Experimental Results

In the following training results, we used NTK parameterization for the linear layers in the MLP. We emphasize that this choice has little effect on the training and convergence in this case, compared to standard initialization.

In figure 10, we showed empirically that the critical exponent of partial Jacobians are vanished for erf with LayerNorm.

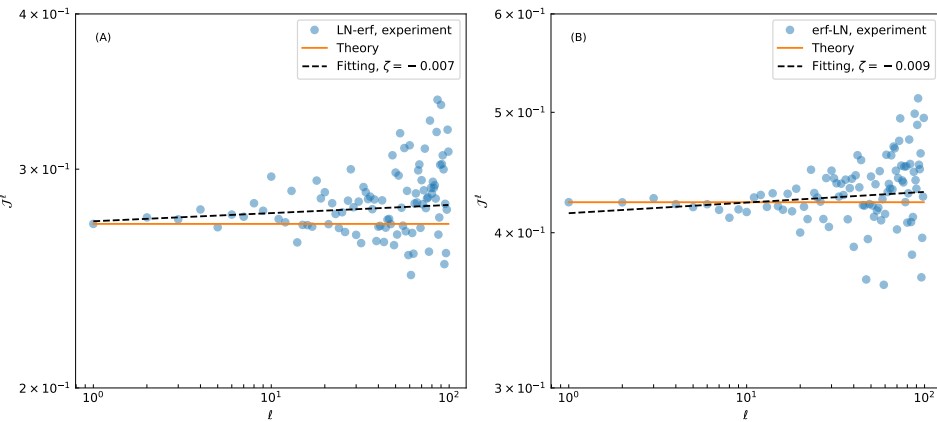

Figure 10: $\log - \log$ plot of partial Jacobian $\mathcal{J}^{0,l}$ vs. $l$ for (A) LN-erf and (B) erf-LN.

In figure 11, we tested $6k$ samples from CIFAR-10 dataset[28] with kernel regression based on neural tangents library [40] [31] [41]. Test accuracy from kernel regression reflects the trainability (training accuracy) with SGD in the ordered phase. We found that the trainable depth is predicted by the correlation length $c\xi$ with LayerNorm applied to preactivations, where the prefactor $c = 28$. The prefactor we had is the same as vanilla cases in [55]. The difference is from the fact that they used $\log_{10}$ and we used $\log_e$.

In figure 12, we explore the broad range in $\sigma_w^2$ of the performance of MLP network with erf activation function and LayerNorm on preativations. The network has depth $L = 50$ and width $N_l = 500$; and is trained using SGD on Fashion MNIST. The learning rates are chosen based on a logarithmic scan with a short training time.

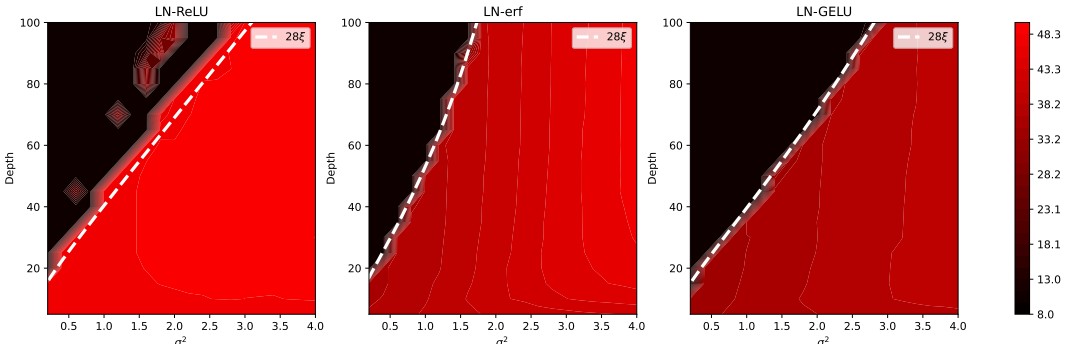

Figure 11: Test accuracy for LayerNorm applied to preactivations. $\sigma_b^2 = 0.5$ for all cases. Correlation lengths are calculated using analytical results of $\chi_{\mathcal{J}}^l$.

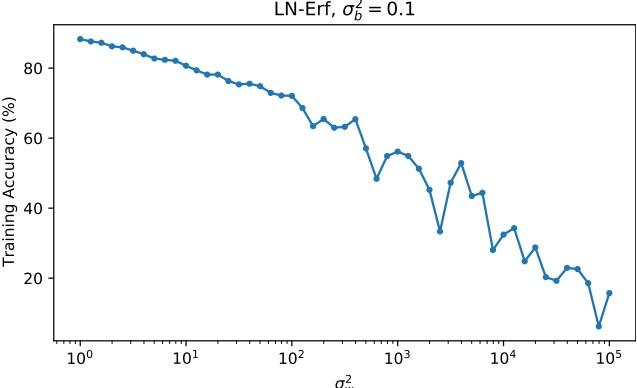

Figure 12: Training performance of MLP networks with erf activation function; and LayerNorm applied to preactivations. It continues to train for several orders of magnitude of $\sigma_w^2$ (with learning-rate tuning).

