# OpenReview forum: "Critical Initialization of Wide and Deep Neural Networks using Partial Jacobians: General Theory and Applications"
_NeurIPS.cc/2023/Conference — NeurIPS 2023 spotlight_

### Official Review · Reviewer_NnYC · 2023-06-23

**Soundness:** 3 good
**Presentation:** 2 fair
**Contribution:** 3 good
**Rating:** 6
**Confidence:** 3

**Summary:**

This paper studies criticality of deep neural networks at initialization. The authors propose a new practical way to diagnose criticality by introducing the partial Jacobian of the network and analyzing the averaged partial Jacobian norm (APJN) and its recurrence relation at large depth. The authors then apply their method to analyze criticality in fully connected networks with LayerNorm and/or residual connections, providing theoretical analysis for infinitely wide networks and a numerical test to select optimal initialization and identifying conditions on network architecture that allow for criticality for any initialization.

**Strengths:**

1. The paper presents theoretical analysis of the APJN in the infinite width limit for various network architectures, and thorough validation by numerics.
2. The paper extensively explored application of the theoretical analysis and numerical test on modern architectures (ResNet and MLPmixer), providing practical insights for initializing neural networks for improved trainability.


**Weaknesses:**

1. As most other works studying critical initialization, the work focuses on networks at Initialization with Gaussian random weights, and there is no learning in the network. While the authors briefly mentioned NTK (line 150), I would like to see a discussion on how their approach may be extended for analysis beyond networks at initialization and perhaps shed light on the learning dynamics of the network.
2. In section 1.2, the authors discussed several related works and how part of their results were previously obtained in a different form in these works. I would recommend that the authors further stress the novelty of their approach/analysis in this section.


**Questions:**

1. How is the APJN defined in this paper related to the order parameters used in previous works [Poole et al, Schoenholz et al.] for criticality (specifically the $c^*$ and $\chi_1$ defined in Eq.4-5 in [Schoenholz et al.]). It would be nice if the authors can add a paragraph on how the APJN is related to these order parameters, and how the empirical test they introduced would be superior to evaluating scalings of the order parameters directly (providing a simpler numerical test? Unbounded activation functions?).
2. Line 146: I am a bit confused what are the respective roles of $\chi_K$ and $\chi_J$ in the trainability of the network and how they are related to each other and the conditions on $\chi_1$ and $c^*$  in [Poole et al.], can you clarify in this paragraph?
3. Section 2: There are two limits here, the infinite width and the infinite depth. It seems that it is important for the scope of this paper to take the infinite width limit (successively) first, is that correct?
4. Figure 3: This figure perhaps need a better caption. Are the white dashed lines given by the infinite width analysis? Why does there seem to be a larger discrepancy in networks with LayerNorm?


**Limitations:**

The authors mentioned in various parts of the paper but did not specifically summarize the limitations, I would recommend adding a paragraph in the conclusion section.

---

> ### Author Rebuttal · Authors · 2023-08-08
>
> We thank the reviewer for the constructive feedback.
>
> ## Training Dynamics and Relation to NTK
>
> The training regime that we target enjoys feature learning and large learning rates, and hence, is far from the NTK regime. While it is possible to connect APJN with NTK and analyze linear dynamics, that is not the purpose of our work. (We can readily add the quantitative relation between APJN and NTK in the Appendix.) Moreover, dynamics beyond the NTK regime for non-linear networks with $L>2$ is a famously unsolved problem.
>
> ## Novelty
>
> The reviewer has correctly identified the key advantages of our work. We elaborate them here:
>
> - All previous diagnostics of criticality require either closed-form solutions or an integral formula followed by numerical estimation [1,2]. This limits their usage to general/unseen architectures. Empirical APJN circumvents these limitations, and is useful as a numerical test for *general* real-world architectures. \
> In addition to the architectures presented in the text, one can analyze inhomogeneous networks by dividing them into "blocks" of layers and considering block-to-block Jacobians.
>
> - $\chi_{\mathcal J}$ can be numerically calculated with a single layer-to-layer gradient, which is readily obtained using Autodiff. Consequently, our method is simple and computationally cheap.
>
> - The characterization of everywhere-critical regime using normalization layers and residual connections. This finding, to our knowledge, is new.
>
> In Section 1.2 of the final version, we will emphasize our contributions and the novelty of our approach.
>
> ## The role of $\chi_{\mathcal J}, \chi_{\mathcal K}$ and relation to $c^*, \chi_1$:
>
> $\chi_{\mathcal J}^l$ is defined as the layer-to-layer APJN $\mathcal J^{l,l+1}$. For large enough $l$ it converges to the fixed point $\chi_{\mathcal J}^*$. The constraint $\chi_{\mathcal J} = 1$ regulates the backward pass (gradient propagation) in the network. For Fully connected networks (FCNs), $\chi_\mathcal J^*$ can be viewed as a generalization of $\chi_1$ defined in the works [1,2]. The key advantages of our method over theirs are:
>
> - Unlike $\chi_1$, $\chi_{\mathcal J}^*$ can be applied to both bounded and unbounded activation functions. (Note that $c^*$ in the aforementioned works fails to capture the critical behavior for unbounded activations such as ReLU -- $c^*=1$ in both phases.) We point this out in line 148.
>
> - As mentioned in the "Novelty" section above, empirical $\chi_{\mathcal J}$ is computationally cheap and has general applicability compared to the aforementioned works.
>
> While $\chi_{\mathcal J}=1$ regulates the backward pass of the network (i.e. gradient updates), $\chi_{\mathcal K}=1$ regulates the forward pass of the networks (i.e. preactivation norms). In practice, we find that the condition on $\chi_{\mathcal J}$ is more important for trainability.
>
> We will state these points more clearly in the final version.
>
> ## Order of limits in Section 2
>
> Yes, for the scope of our work, it is important to take the width to infinity first. More precisely, we take the successive layer widths to infinity (sequential limit in Theorem 1.3). In practice, this translates to $L/N \sim o(1)$.
>
> ## Figure 3 Caption and Clarifications
>
> We had to truncate the image captions due to space constraints. In the final version, we will add the following caption for Figure 3:
>
> "Figure 3: Trainability (Training accuracy) of deep MLP on FashionMNIST $(N_l=500, L=50)$. The dotted white line denotes the (analytical) critical lines. Note that the combination of LayerNorm and residual connections significantly improve trainability, with the $\mu=1$ case being everywhere-trainable."
>
> Yes, the white dashed lines are critical lines from theory.
>
> We are unsure of what the reviewer means by "larger discrepancy in networks with LayerNorm". We answer this question with two possible interpretations. We urge the reviewer to clarify their question so we can better answer it.
>
> - Networks with LayerNorm seem trainable even far from the critical line (dotted white line): LayerNorm, especially in conjunction with residual connections drastically improves the correlation length. For real, finite-depth networks, correlation length sets the scale for trainable depth. This results in finite-depth networks training well in a finite region around criticality.
>
> - Critical lines have seemingly different slopes in Figures 2 and 3: This is simply the result of the span of the Y axes of the plots. They are identical otherwise.
>
> [1] B. Poole, et al., Exponential expressivity in deep neural networks through transient chaos, 2016.
>
> [2] S. S. Schoenholz, Deep Information Propagation, 2016.

---

### Official Review · Reviewer_8MUF · 2023-07-05

**Soundness:** 3 good
**Presentation:** 4 excellent
**Contribution:** 3 good
**Rating:** 7
**Confidence:** 3

**Summary:**

The paper addresses the theoretical treatment of deep neural networks and introduces a novel practical approach to identify criticality within these networks. The authors work in the setting where the number of parameters per layer approaches infinity, enabling the formulation of quantitatively predictive descriptions ( establish criteria for hyperparameter selection). These criteria are based on the notion of criticality.¨

To identify criticality, the paper introduces partial Jacobians, which represent derivatives of preactivations in layer l with respect to preactivations in layer l0 (where l0 ≤ l). Recurrence relations for the norms of these partial Jacobians are derived and utilized to analyze criticality in deep fully connected neural networks featuring LayerNorm and/or residual connections.

The authors devise a straightforward and cost-effective numerical test to determine the optimal initialization for various types of deep neural networks, including fully connected, convolutional, and normalization layers. They present quantitative evidence demonstrating that arranging LayerNorm (applied to preactivations) and residual connections appropriately leads to an architecture that exhibits criticality regardless of the initialization.

Finally, the paper applies these methods to investigate the ResNet and MLP-Mixer architectures, revealing the presence of an everywhere-critical regime within these modern models.

**Strengths:**

This research represents a significant advancement in the field. A notable contribution of the paper is the introduction of partial Jacobians and their averaged norms as powerful tools for analyzing gradient propagation in deep neural networks at initialization. The paper particularly investigates the implications of LayerNorm and residual connections, shedding light on their impact on trainability.
 It is encouraging that the theoretical prediction derived from this Framework match previously empirically observed or theoretically proven findings. Additionally, the paper strengthens its contributions by testing the findings on more realistic datasets, thereby enhancing the robustness and applicability of the research.

The paper is highly comprehensive and lucid, greatly facilitating understanding. It effectively summarizes the contributions, allowing readers to grasp the main points without constant reference to the appendix. Furthermore, the authors have shared the relevant code with their submission, ensuring transparency and enabling the replication of results. The inclusion of detailed step-by-step derivations significantly aids comprehension and is highly appreciated by readers.


**Weaknesses:**

These suggestions are intended to improve the overall clarity and accessibility of the research.

To enhance the organization of the paper, it would be beneficial to include dedicated sections for limitations, future work and contributions. Additionally, the current title of the section labeled "results" may be misleading and should be reconsidered. Furthermore, FashonMNIST is not mentioned in the main text.

The limitations outlined below highlight some of the weaknesses of the paper, despite the fact that they represent the current best efforts in the field.

Although it may seem like a minor detail, I believe it holds significant importance. For the sake of accessibility, completeness, and readability, I would suggest including relevant lemmas or theorems used from other sources in the appendix to provide a more comprehensive understanding of the research.  Furthermore, It would be helpful if you could extend slightly the captions in the figures for the camera-ready version for the same reason.



**Questions:**

Dear authors, I have a few questions regarding your work that would greatly aid my comprehension:

* Can you provide a rationale for selecting the architectures used in your study? Are there any other potential architectures that could have been considered within the framework?

* In addition to LayerNorm and residual connection what other technics used in the field could have been explored in your framework ?

Thank you in advance.

**Limitations:**

To enhance the paper, it would be valuable to provide a clear outline of the limitations of the work. Some potential limitations to consider include:

The paper focuses on testing a specific set of realistic architectures, which means there is room for exploring other architectures beyond those examined in the study. Although the obtained results are encouraging, extending the analysis to different architectures would provide a more comprehensive understanding of the findings.

Additionally, the paper focuses on investigating the effects of LayerNorm and residual connections. While these factors are certainly important, exploring the impact of other components or techniques apart from LayerNorm and recurrent layers could broaden the scope of the research. Examining different elements in the network architecture could reveal additional insights into their contributions and interactions.

Another important limitation of the paper is that it specifically focuses on the infinite width limit of deep neural networks. While analyzing neural networks in the infinite width limit can provide valuable theoretical insights, it may not fully capture the behavior and characteristics of networks with finite widths. To address this limitation, it would be valuable to extend the analysis to include investigations and experiments on networks with various finite widths. This would provide a more realistic and comprehensive understanding of the behavior, performance, and criticality of deep neural networks in practical settings.

Including a note on the computational complexity required to conduct these experiments would be beneficial. Acknowledging the computational demands of the experiments would provide a better understanding of the resources needed for replication or further research.

---

> ### Author Rebuttal · Authors · 2023-08-08
>
> We thank the reviewer for their encouraging feedback and suggestions for improvement.
>
> ## Presentation and Clarity
>
> We welcome the suggestions to improve the presentation of our paper:
> - We will include a discussion on limitations and future work in Section 7 (Conclusion).
>
> - We will change the Section title to "Main Result", as it states "everywhere-criticality", a key contribution of our work.
>
> - We mention FashionMNIST in the caption for Figure 3.
>
> - We will extend the figure captions (especially Figure 3) in the final version of the paper. (See Global Response for further discussion.)
>
> ## Answers to Questions:
>
> 1. The applicability of *numerical* Averaged Patial Jacobian Norm (APJN) is quite general. The choice of modern architectures like ResNet and MLP-Mixer was made to demonstrate its utility in SOTA models. It is possible to extend this method to models with Attention layers (eg. Transformers). We find that APJN correctly predicts the everywhere-criticality of pre-LN Transformer. For further details, we refer the reviewer to the General Response and the accompanying one-page pdf.
>
> 2. As we mention in the paper, the analysis also extends to other normalization techniques such as BatchNorm and GroupNorm. Dropout can also be included. In general, one can divide the network into "blocks" of layers and study APJN wrt outputs of these blocks. Since these blocks can contain arbitrary layers/techniques within them, any/all models with this "feedforward" structure fall within the domain of applicability of APJN.
>
> ## Infinite Width Limit
>
> We would like to emphasize that although our theoretical analyses are performed in infinite width limit, all training (including MLP, ResNet and MLP-Mixer) are performed on real, finite-width networks. For example, all MLP experiments were performed with N=500, L=50.
>
> Formally, the notion of criticality remains crisp for networks with $L/N \sim o(1)$. $L/N$ corrections to the infinite width analysis have been calculated in other works [1][2]. Finite $L/N$ creates fluctuations in partial Jacobian-norm around its mean value. Nevertheless, for practical purposes, the notion of criticality remains largely unchanged. This can be readily seen by the agreement of our finite-width experiments (phase diagrams) and infinite-width predictions (dashed lines) in Figure 2.
>
> We direct the reviewer to Global Response for further remarks.
>
> ## Extension to other architectures and techniques
>
> - As mentioned above, empirical application of APJN is quite general. On the theoretical side, a comprehensive extension of our methods to Transformers would be very useful; especially due to the ubiquity of Transformers across various tasks/modalities.
>
> - We study normalization techniques like LayerNorm as well as residual connections because of their prevalence in modern architectures. We encourage further studies of our methods on domain-specific techniques.
>
> ## Computational Resources
>
> Appendix A outlines the computational resources utilized in performing each of our experiments. In general, the computation of empirical APJN takes $<1\\%$ of the resources required for training. We are happy to provide further details.
>
> [1] S. Yaida, Non-Gaussian processes and neural networks at finite widths, 2020.
>
> [2] B. Hanin, Random Fully Connected Neural Networks as Perturbatively Solvable Hierarchies, 2022.

---

> > ### Comment · Reviewer_8MUF · 2023-08-12
> >
> > Many thanks to the authors for your careful explanation and detailed rebuttal. I increased my score as a reflection of improvement in clarity and presentation as well as the general understanding of the contribution.

---

> > > ### Author Response · Authors · 2023-08-20
> > >
> > > We thank the reviewer for their response and the careful consideration of our manuscript. We are glad to have addressed the reviewer's questions and concerns satisfactorily; and are grateful for the score revision. We are happy to answer any further questions.

---

### Official Review · Reviewer_EkqX · 2023-07-06

**Soundness:** 3 good
**Presentation:** 2 fair
**Contribution:** 3 good
**Rating:** 6
**Confidence:** 4

**Summary:**

The paper studies the effect of the expected value of the Jacobian norm of a particular layer with respect to a previous layer as the depth of a neural network (NNs) increases. The study is done under the assumption of infinite width NNs, and with the goal of assessing sensitivity to initialisation hyperparameters (the standard deviation of the normal initialiser of the weight and biases of the network) depending on architecture and as depth increases. The importance of understanding this setting comes from the connection with exploding and vanishing gradients that the network is likely to exhibit in training, unless the initialisation (in expectation) is in the critical region. The conditions for criticality across architectures are studied in this work.

The settings studied include feed-forward networks, residual networks with and without layer-norm applied to the pre-activation, as well as group norm and briefly Batch Norm in conjunction with residual networks.

Reasons for rating: While the contribution of the paper is worthwhile, the clarity could be greatly improved. At the moment, the paper resembles more a series of facts rather than a clear logical deduction. I worry that this will limit the value it can bring to the ML community.


**Strengths:**

Strengths:
 * The paper provides a theoretical assessment of why, in the infinite width limit, certain neural networks can achieve better trainability. In particular, the authors study the effect of residual connections and layer norm, both which have become a staple of deep learning. This type of study can help explain why certain architectures are less sensitive to initialisation compared to others.
 * Code reproducing the figures is provided in the supplementary material as notebooks (and a clear readme.txt detailing the results).
 * Proofs in the attached SM are generally readable and can be followed.


**Weaknesses:**

Weaknesses:
* The clarity of the paper can be drastically improved.
  * In the paper, the same recipe of proof is applied repeatedly in multiple settings. I think the paper would benefit from clarifying that recipe in the main text, and working through the logical steps one by one in one example. At the moment, there is no clear explanation of the logical flow of what is occurring. My understanding of that recipe is:  In the setting studied (for that specific architecture), in the infinite width limit find a recurrence relationship between the NNGP kernels at layer l+1 and l when evaluated at the same datapoint. Use that recurrence relationship, together with the recurrence relationship for APJN to find the conditions of criticality for initialisations. I urge the authors to clarify this recipe (in more detail) in the main manuscript. as well as work through an example.
 * There are many sentences for which it is unclear where they come from, either from a proof in the Appendix or from a previous work or the authors consider it trivial. Such an example is Eq 4, which potentially follows from Thm 1.3, but the order of the two is unclear. Similarly, the proof of Thm 1.3 in the Appendix is not well delimited or clarified (given that it is the main result).


**Questions:**

Suggestions for improvement:
 * Please see above in terms of weaknesses.
  * For Thm 2.4 (and similarly in the Appendix proofs), why not write the recurrence relationship as K^{(l+1)}(x, x) = \sigma_w^2 K^{(l)}(x, x) +  \sigma_b^2 ?
  * There are many sentences in the paper that require additional citations, such as line 213.
 * Figure 1b shows a lot of noise as the number of layers increases, can the authors comment on that?
* The authors should provide additional details on how the figures are obtained, at the moment these are scant.


**Limitations:**

I believe that the biggest limitation of the paper as is now is its presentation, which can be drastically improved. This applies both to the writing and the details around the figures.

---

> ### Author Rebuttal · Authors · 2023-08-08
>
> We thank the reviewer for the constructive feedback and detailed suggestions on the presentation.
>
> ## Clarity
>
> We believe that the logical flow of the current manuscript is similar to what the reviewer has proposed. We state the outline of our paper here -- we will add a version of this in the Introduction of the final version:
>
> In section 2 we introduce the recurrence relation for NNGP kernel and Jacobians with fully connected networks. Based on these quantities we define criticality. In Section 3 and Section 4, we add LayerNorm and residual connections, describing their effects on criticality. Then in Section 5, we combine LayerNorm and residual connections to show the emergence of everywhere-criticaly. Finally, we use our methods on real-world architectures like ResNet and MLP-Mixer in Section 6.
>
> ## Proof of Eq 4
>
> We use Section 1.1 to summarize our theoretical results. Eq. 4 can be viewed as the definition of correlation length $\xi$. Theorem 1.3 then utilizes this definition and states the effect of LayerNorm and residual connections on correlation length.
> An additional explanation for Eq. 4 is provided in Section 2.1. In the final version, we will add a reference to Section 2.1 as a motivation for Eq 4.
>
> ## Proof of Thm 1.3
>
> Theorem 1.3 directly follows from Eq. 77 (Appendix F), upon taking infinite width and large depth limits. (and taking the logarithm, as mentioned in the text below Eq 77). We will flesh out these steps more explicitly in the final version.
>
> ## Form of recursion equations:
>
> We offer multiple responses to 2 possible interpretations of the reviewer's question:
>
> (i) If the reviewer is suggesting the specific form: The form that the reviewer mentioned, $K^{(l+1)}(x, x) = \sigma_w^2 K^{(l)}(x, x) + \sigma_b^2$, only holds for linear networks. For other cases $K^{(l+1)}(x, x)$ is a non-linear map of $K^{(l)}(x, x)$, which depends on the activation functions.
>
> (ii) If the reviewer means to ask why we keep the summation and the factors of $1 / N_l$ in the equations: It is possible to introduce some extra symbols to simplify the equations notationally. However, we believe that such additional notation will come at the price of clarity and lucidity. The current way of writing formulae makes the underlying calculations apparent. This should make the equations accessible to a broader audience.
>
> ## Additional references:
>
> We thank the reviewer for pointing out places where more references are potentially warranted. We will scan the text and add the relevant references.
>
> ## Noise in Figure 1(b)
>
> The apparent fluctuations in Figure 1(b) come from two factors:
>
> (i) The Y-scale of Figure 1(b) is more zoomed in compared to other plots, making the fluctuations appear strong. We chose this scale because the underlying curve of $\mathcal J^{0, l}$ is a constant -- without increase or decrease in the trend, fluctuation offers the only natural choice for the scale of the y-axis.
>
> (ii) The existence of the fluctuations is a result of the depth-to-width ratio ($L/N$). $L/N = 250/1000 = 0.25$ in this case. This results in strong fluctuations even after averaging.
>
> ## Details of figures
>
> The experimental details of all the figures, along with computational resources for reproducing them, are fleshed out in Appendix A (due to space limitations). In the final version, we will expand the figure-captions by adding further experimental details.

---

> > ### Comment · Reviewer_EkqX · 2023-08-11
> > **Rebuttal update**
> >
> > I thank the reviewers for the update (and for sharing the empirical transformer results, which while not requested by myself, are nonetheless interesting).
> >
> > I have decided to keep my score. Other reviewers have also highlighted the need for an increase in clarity of the paper, and I do not feel like the above changes suggested by the authors above are meaningful enough to greatly increase clarity.

---

> > > ### Author Response · Authors · 2023-08-20
> > > **Concerns about clarity**
> > >
> > > We thank the reviewer for their response and careful consideration of our manuscript.
> > >
> > > To further address the reviewer's concerns about clarity, we will add a summary of the recipe for the derivation of recurrence relations at the beginning of Section 2. To that end, we will include the following paragraph above Definition 2.1:
> > >
> > > ``Here we derive the infinite width recurrence relations for the APJN and the NNGP kernel. We use Lemma 2.2 to derive NNGP kernel recursion, and leverage that to get the recursion for APJN. Fixed point analyses of these relations help us define critical line and point. We discuss the vanilla MLP with no LayerNorm and $\mu=0$ in this section. Results with LayerNorm and/or residual connections as well as modern architectures are presented in the following sections. (We refer the reader to Appendices C,E,F for proofs and detailed application of the above recipe. Appendices H,I,J contain the results for various activation functions.)''
> > >
> > > We will also modify Appendices C,E,F to make the flow of proofs more apparent. Due to space constraints, we reserve the detailed proofs and various example settings (LayerNorm, residual connections etc.) to Appendices C,E,F; and example activation functions to Appendices H,I,J.
> > >
> > > We hope to have satisfactorily addressed the reviewer's concerns about the presentation. We welcome any further questions and suggestions.

---

### Official Review · Reviewer_RvQ9 · 2023-07-06

**Soundness:** 4 excellent
**Presentation:** 3 good
**Contribution:** 3 good
**Rating:** 7
**Confidence:** 3

**Summary:**

The paper presents a theoretical framework for understanding the trainability of deep neural networks with LayerNorm and residual connections. The authors derive analytical expressions for the neural network Gaussian process (NNGP) kernel and the partial Jacobian norm (PJN) for a wide range of activation functions. They show that the combination of LayerNorm and residual connections leads to an everywhere-critical regime, where the network can be trained effectively irrespective of the initialization. The authors also provide insights into the role of the hyperparameters and the activation function in determining the trainability of the network. The paper's contributions include a theoretical understanding of the trainability of deep neural networks with LayerNorm and residual connections and insights into the design of effective architectures.

**Strengths:**

The main strengths of the paper are:
1. Introduces partial Jacobians and their averaged norms as tools to analyze the propagation of gradients through deep neural networks at initialization.
2. Presents a very cheap and simple empirical test for criticality using APJN evaluated close to the output.
3. Shows that criticality formulated in terms of partial Jacobians is equivalent to criticality studied previously in the literature.
4. Investigates homogeneous architectures that include fully-connected layers, normalization layers, and residual connections, and shows that the combination of LayerNorm and residual connections can drastically increase correlation length leading to improved trainability.
5. Considers examples of modern architectures, ResNet and MLP-Mixer, and shows that they are critical everywhere at µ = 1 (i.e., with residual connections) due to the interaction between LayerNorm and residual connections.
6. Empirically demonstrates that deep MLP-Mixer with µ = 1 trains well for various initializations.

**Weaknesses:**

Some potential limitations of the paper could include:
1. The analysis is limited to homogeneous architectures with fully-connected layers, normalization layers, and residual connections. The results may not generalize to other types of architectures or layers (e.g., attention layers).
2. The paper focuses on the infinite width limit, which may not be directly applicable to finite-width networks commonly used in practice.
3. The empirical test for criticality based on the averaged partial Jacobian norm may not be sufficient to fully capture the behavior of deep neural networks (in terms of accuracy for instance).
4. The paper does not provide a comprehensive comparison with other methods for improving the trainability of deep neural networks, such as weight initialization schemes or adaptive optimization algorithms.

**Questions:**

See the weaknesses part.

**Limitations:**

See the weaknesses part.

---

> ### Author Rebuttal · Authors · 2023-08-08
>
> We thank the reviewer for the constructive feedback.
>
> ## Infinite width limit
>
> Formally, our theoretical results become crisp when $L/N \sim o(1)$ (width $N$, depth $L$). In practice, our conclusions apply whenever this ratio is small. For instance, all the phase diagrams in figures 2,4,5 are obtained on real, finite width, deep networks (MLP: N=500, L=50; ResNet and MLP-Mixer: N=512, L=32). One can readily see the agreement between finite-width phase diagrams and infinite-width predictions (dashed lines).
>
> Moreover, in practice, very deep architectures typically use residual connections. Residual connections make networks effectively shallow, leading to an effectively small ratio $L/N$.
>
> We direct the reviewer to "Global Response" for more details.
>
> ## Inhomogeneous architectures
>
> The utility of Averaged Partial Jacobian Norm (APJN) can be readily generalized to inhomogeneous architectures. These architectures can contain differently initialized fully-connected layers, convolutional layers, normalization layers, residual connections, and even attention layers. In these case, due to inhomogeneity, the fixed point analysis and scaling discussion no longer applies. However, it is possible to impose criticality with a stricter requirement: $\mathcal J^{l,l+1}=1$ for all layers. This requirement fixes the hyperparameters layer-by-layer, which can be implemented using our code. The additional coputational cost required in this case is no more than other methods for critical initializations. Though more computational power is required in this case, we view this as a price one must pay no matter which methods one uses.
>
> ## Attention
>
> We present empirical phase diagrams for the Vision Transformer (ViT) with various settings, in the accompanying on-page pdf. Our empirical methods correctly measure the gradient scale, including the fact that pre-LN transformers are better behaved than post-LN transformers.
>
> On the theoretical side, the analysis is more involved. Attention layers can be viewed as fully-connected layers with data-dependent weights. It is known that the commonly used scalar-dot product attention layer converges to a Gaussian process in the limit where the number of heads and the embedding dimension go to infinity [1]. For most real-word models $L / n_{heads}$ is $O(1)$, which makes infinite width analysis unfavorable even with residual connections. Furthermore, softmax function in the attention prevents getting closed-form results. Despite the lack of proper theoretical understanding, our tools can remarkably analyse criticality for such networks (See attached one-page PDF).
>
> ## Accuracy
>
> The objective of our methods is not to predict the best performance based on initialization. In general, it is hard to isolate the effect of initialization from factors like dataset, optimization, regularization etc. Our goal is to provide a universal framework for initializing a network that facilitates good training performance. Such a framework is invaluable, for instance, in Neural Architecture Search (NAS); where it is important to distinguish the effect of architecture from initialization. Our experiments show that APJN manages to achieve this in a diverse set of cases.
>
> ## Comparison
>
> We will add more related works and explain the difference and connections between known methods and ours. Here we just want to summarize the key point briefly:
>
> - Popular initialization schemes, for example, He initialization, is a special case of our method. These schemes apply to specific cases (e.g. He init is critical for ReLU networks).
>
> - References to prior methods that utilize the infinite width limit to fix initializations (e.g. $\sigma_w^2$) are included in lines 146-150.
>
> - Methods focusing on scaling, including Fix-up[2], ReZero[3], and other works, those methods can also be summarized as initialize with $\mathcal J^{l, l+1} = 1$.
>
> - Good initialization is essential even with adaptive methods and should be used in conjunction. For Adam, at early training time, the gradient update is bounded by the learning rate $\eta$ [4][5]. At layer $l$, the difference between the feature-updates of two adjacent input vectors $x$ and $x+\epsilon$ is upper bounded by $O(\eta \cdot \epsilon \cdot \mathcal J^{0, l})$ (up to some architecture/loss dependent constant). In this case, $\mathcal J^{0, l}$ regulates the relative updates for different features, preventing them from being too large or too small, facilitating better training.
>
> [1] J. Hron, et al., Infinite attention: NNGP and NTK for deep attention networks, 2020.
>
> [2] H. Zhang, Y. N. Dauphin, T. Ma, Fixup Initialization: Residual Learning Without Normalization, 2020.
>
> [3] T. Bachlechner, et al., ReZero is All You Need: Fast Convergence at Large Depth, 2020.
>
> [4] D. P. Kingma, J. Ba, Adam: A Method for Stochastic Optimization, 2017.
>
> [5] J, Ma, D. Yarats, On the Adequacy of Untuned Warmup for Adaptive Optimization, 2021.

---

### Author Rebuttal · Authors · 2023-08-08

# Global Response to All Reviewers

## Infinite width limit

For analytical results, we first take the infinite width limit and then a large depth limit. Stated differently, it means that the depth-to-width ratio $L/N \sim o(1)$. In practice, this assumption holds as long as $L/N$ is small. Moreover, residual connections make networks effectively shallow, significantly decreasing the effective $L/N$ ratio, and allowing our conclusions to be applied at even larger depths.

All of our phase diagrams and training results are obtained from real networks with reasonable width and depth (MLP: N=500, L=50; ResNet and MLP-Mixer: N=512, L=32). This should convince the reviewers that although our theory is formulated in the infinite width limit, our results concerning criticality describe real-world models; albeit with an error (fluctuation) of up to $O(L/N)$.

## Transformer Architectures

We have added phase diagrams for Vision Transformers (ViTs) in the global response PDF file. As predicted, $\mu=1$ with pre-LN is everywhere critical. Removing LayerNorm or using smaller $\mu$ leads to non-critical initializations. We will add these results to the Appendix of the final version.

On the theoretical side, the analysis is more involved. It is known that the commonly used scalar-dot product attention layer converges to a Gaussian process in the limit where the number of heads and the embedding dimensions go to infinity [1]. However, real-world models are far from this limit; wherein $L/n_{heads}$ is $O(1)$ (even with residual connections). This makes infinite-width analysis unfavorable. Moreover, Softmax function in attention prevents closed-form results for most calculations. Despite these limitations, people have found a particular width scaling of initialization and learning rate for pre-LN transformers [2].

Despite the lack of comprehensive theoretical understanding, our tools can extract meaningful information about the magnitude of gradients for attention layers (See attached one-page pdf). It identifies criticality and correctly predicts that pre-LN transformers are better-behaved than post-LN transformers [3].

## Captions and Clarity

Due to space limitations, we had oversimplified some of the Figure-captions. We will write more detailed captions in the final version. Details of experiments and computational resources are fleshed out in Appendix A.

[1] J. Hron, et al., Infinite attention: NNGP and NTK for deep attention networks, 2020.

[2] E. Dinan, et al., Effective Theory of Transformers at Initialization, 2023.

[3] R. Xiong, et al., On Layer Normalization in the Transformer Architecture, 2020.

---

### Decision · Program_Chairs · 2023-09-21

**Decision:**

Accept (spotlight)

**Comment:**

The paper introduces a novel practical approach to identify criticality within neural networks. Using the infinite width limit, the authors derive analytical expressions for the neural network Gaussian process kernel and the partial Jacobian norm for a wide range of settings. The authors show that the combination of LayerNorm and residual connections leads to an everywhere-critical regime, where the network can be trained effectively irrespective of the initialization. The paper's contributions include a theoretical understanding of the trainability of deep neural networks with LayerNorm and residual connections via averaged partial Jacobian Norm and insights into the design of effective architectures. The authors apply these methods to investigate the ResNet and MLP-Mixer architectures, revealing the presence of an everywhere-critical regime within these modern settings.


[Strength highlighted by reviewers]

- Introduces partial Jacobians and their averaged norms as tools to analyze the propagation of gradients through deep neural networks at initialization
- The paper presents theoretical analysis of the APJN in the infinite width limit for various network architectures, and thorough validation by numerics.
- Presents a computationally cheap and simple empirical test for criticality using APJN evaluated close to the output.
- Shows that criticality formulated in terms of partial Jacobians is equivalent to criticality studied previously in the literature.
- Investigates homogeneous architectures that include fully-connected layers, normalization layers, and residual connections, and shows that the combination of LayerNorm and residual connections can drastically increase correlation length leading to improved trainability.
- The paper explored application of the theoretical analysis and numerical test on modern architectures (ResNet and MLPmixer), providing practical insights  that they are critical everywhere at µ = 1 (i.e., with residual connections) due to the interaction between LayerNorm and residual connections.
- The paper provides a theoretical assessment of why, in the infinite width limit, certain neural networks can achieve better trainability. In particular, the authors study the effect of residual connections and layer norm, both which have become a staple of deep learning. This type of study can help explain why certain architectures are less sensitive to initialisation compared to others.
- Code reproducing the figures is provided in the supplementary material as notebooks

[Weakness highlighted by reviewers] Main issues raised by reviewers are on clarity and suggested many forms / directions suggesting organizing presentations based on clear logical deduction.

[Overall] After the rebuttal, all reviewers supported accepting the paper (score of 7, 7, 6, 6) saying "the contribution of the paper is worthwhile", " a significant advancement in the field", "powerful tools for analyzing gradient propagation in deep neural networks at initialization", "paper is highly comprehensive and lucid, greatly facilitating understanding". While there are still issues on clarity remaining after rebuttal, the AC believes this can be effectively addressed if authors incorporate many promised changes suggested by the reviewers. I strongly encourage the authors to take these changes seriously to improve clarity and presentation to help the readers.